# Thinking-Free Policy Initialization Makes Distilled Reasoning Models More Effective and Efficient Reasoners

**Xin Xu**[1,2], **Clive Bai**[2], **Kai Yang**[2], **Tianhao Chen**[1], **Yang Wang**[5], **Saiyong Yang**[2,†], **Can Yang**[1,3,4,†]

[1]Department of Mathematics, HKUST [2]HY, Tencent
[3]Big Data Bio-Intelligence Lab, HKUST [4]State Key Laboratory of Nervous System Disorders, HKUST
[5]Department of Mathematics, HKU
 **Github Repo**: [GitHub Page]  **Models**: [Huggingface Models]

## Abstract

Reinforcement Learning with Verifiable Reward (RLVR) effectively solves complex tasks but demands extremely long context lengths during training, leading to substantial computational costs. While multi-stage training can partially mitigate this, starting with overly short contexts often causes irreversible performance degradation, ultimately failing to reduce overall training compute significantly. In this paper, we introduce **T**hinking-**F**ree **P**olicy **I**nitialization (**TFPI**), a simple yet effective adaptation to RLVR that bridges long Chain-of-Thought (CoT) distillation and standard RLVR. TFPI employs a simple *ThinkingFree* operation, explicitly discarding the thinking content via a direct `</think>` append, to reduce token usage during inference. Training with *ThinkingFree*-adapted inputs improves performance and lowers token consumption, even in the original slow-thinking mode. Extensive experiments across various benchmarks have shown that TFPI accelerates RL convergence, achieves a higher performance ceiling, and yields more token-efficient reasoning models without specialized rewards or complex training designs. With TFPI only, we train a 4B model to reach 89.0% accuracy on AIME24 and 65.5% on LiveCodeBench using less than 4K H20 hours.

## 1 Introduction

Reasoning is a fundamental aspect of human cognition, and equipping artificial intelligence (AI) with strong reasoning capabilities is critical for its deployment and applications (Morris et al., 2023; Huang et al., 2024; Xu et al., 2025c). Progress in pretraining (Shao et al., 2024; Yang et al., 2024; Chen et al., 2025b), supervised fine-tuning (SFT) (Yu et al., 2023; Xu et al., 2024; Tong et al., 2024; Ye et al., 2025; Muennighoff et al., 2025), rigorous evaluation (Rein et al., 2024; Phan et al., 2025; Xu et al., 2025b), reinforcement learning (RL) (Jaech et al., 2024; Guo et al., 2025) has significantly enhanced the reasoning abilities of large language models (LLMs). Among these, RL with verifiable rewards (RLVR) stands out as an effective approach that enables large language models (LLMs) to generate long Chains-of-Thought (CoT) (Wei et al., 2022) spontaneously, and empowers them with unprecedented performance on challenging reasoning tasks. Thus, these RLVR-trained LLMs are termed as long-CoT LLMs, slow-thinking LLMs, or large reasoning models (LRMs).

Compared with initializing from a base LLM, starting from an SFT-distilled LRM typically yields better results and accelerates convergence in RLVR (Guo et al., 2025; An et al., 2025). However, SFT-distilled LRMs often produce excessively long responses during the rollout stage of RLVR, which necessitates a large training context length for RLVR. Using such large training contexts also incurs substantial computational costs. A common mitigation strategy is multistage RLVR, which begins with a relatively "short" context and gradually increases the training length (Luo et al., 2025b; An et al., 2025). Nonetheless, Zeng et al. (2025a) argue that multistage training might cause

---

† Corresponding Authors.

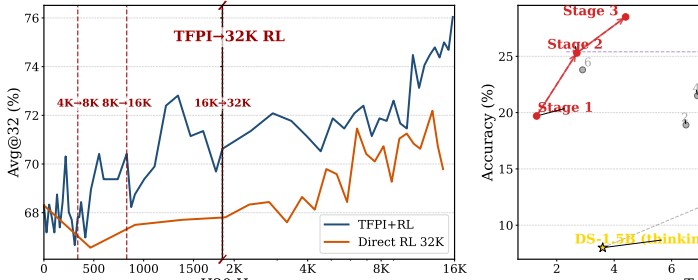

Figure 1: Our proposed TFPI accelerates the convergence of RLVR to a higher performance ceiling (left) and yields more token-efficient reasoning models (right). **Left:** `avg@32` versus training compute, measured in H20 hours. "Direct RL" refers to directly training `Qwen3-4B` with a 32K context window using DAPO, while "TFPI + RL" denotes running 32K-context DAPO after initialization with our 3-stage TFPI. The x-axis for TFPI uses a linear scale during the TFPI phase, followed by a logarithmic scale, with the transition indicated by a black vertical line. **Right:** Average accuracy on 4 reasoning datasets (AIME24/25, Beyond AIME, and GPQA) versus average output tokens. Points in the upper-left region indicate better performance. Baseline names and their corresponding numbers are listed in Table 3. Red dots denote different stages of our TFPI.

irreversible performance degradation. Moreover, even multistage training demands significant computational resources. For instance, training a 4B model while progressively increasing the maximum context length from 40K to 48K to 52K tokens requires approximately 8K H800 GPU hours (An et al., 2025). These limitations underscore the need for more efficient RLVR training methods.

In this work, we introduce **T**hinking-**F**ree **P**olicy **I**nitialization (**TFPI**), a simple yet effective adaptation to RLVR that bridges long Chain-of-Thought (CoT) distillation and standard RLVR. We first observe that a *ThinkingFree* operation—explicitly discarding the thinking content via a direct `</think>` append—can substantially reduce token usage during inference (Section 3.1). Training with *ThinkingFree*-adapted inputs improves performance and lowers token consumption, even when evaluated in the original slow-thinking mode (Section 3.2). We then formally define TFPI in Section 3.3. As illustrated in Figure 1, TFPI accelerates RL convergence, achieves a higher performance ceiling, and produces more token-efficient reasoning models without requiring specialized rewards or complex training designs (Section 4).

Our contributions are as follows: ❶ We find that *ThinkingFree* can substantially reduce inference costs for distilled LRMs, and that training with *ThinkingFree*-adapted inputs enhances slow-thinking capability. ❷ We propose TFPI, a fast, low-cost initialization phase for long-CoT RL that accelerates RL convergence and exhibits transfer across domains, even when trained solely on mathematics. ❸ We show that long-CoT RL following TFPI achieves a higher performance ceiling while producing more token-efficient LRMs without the need for specialized rewards or complex training designs, offering an effective and efficient route to training high-performing LRMs. ❹ We provide both behavioral and parameter-level analyses and reveal that TFPI not only preserves the slow-thinking reasoning pattern but also enables substantially faster rollouts in subsequent long-CoT RL stages.

## 2 PRELIMINARY

**Notation**. In this paper, we define an LLM parameterized by $\theta$ as a policy $\pi_\theta$. Let $x$ denote a query and $\mathcal{D}$ the set of queries. Given a response $y$ to a query $x$, its likelihood under the policy $\pi_\theta$ is expressed as $\pi_\theta(y \mid x) = \prod_{t=1}^{|y|} \pi_\theta(y_t \mid x, y_{<t})$, where $|y|$ denotes the number of tokens in $y$. A query–response pair $(x, y)$ is scored by a rule-based outcome reward $r(x, y) \in \{0, 1\}$, indicating whether the response $y$ aligns with the ground truth of $x$.

**RLVR Algorithms**. RLVR algorithms are central to the recent success of LRMs, with prominent examples including Proximal Policy Optimization (PPO) (Schulman et al., 2017) and Group Relative Policy Optimization (GRPO) (Shao et al., 2024). PPO constrains updates to remain close to the previous policy $\pi_{\theta_{old}}$ via clipping. GRPO bypasses the need for the value model by computing relative advantages among responses to the same query. A widely adopted GRPO variant, DAPO (Yu

et al., 2025), adds clip-higher, dynamic sampling, and token-level policy-gradient losses to stabilize training. The objective is given by $\mathcal{J}_{\text{DAPO}}(\theta) = \mathbb{E}_{x \sim \mathcal{D}}\left[\mathcal{J}_{\text{DAPO}}(\theta, x)\right]$,

$$\mathcal{J}_{\text{DAPO}}(\theta, x) = \left[\frac{1}{\sum_{i=1}^{G}|y_i|}\sum_{i=1}^{G}\sum_{t=1}^{|y_i|}\min\left(r_{i,t}(\theta)\,\widehat{A}_{i,t},\,\text{clip}\left(r_{i,t}(\theta), 1-\varepsilon_{\text{low}}, 1+\varepsilon_{\text{high}}\right)\widehat{A}_{i,t}\right)\right],$$
$$\text{s.t.}\quad 0 < |\{y_i \mid \texttt{is\_equivalent}(y_i, x)\}| < G. \tag{1}$$

Here, $\{y_i\}_{i=1}^{G} \sim \pi_{\theta_{\text{old}}}(\cdot|x)$ and $G$ is the number of generated responses to each query $x$ (i.e., the group size) and the importance ratio $r_{i,t}(\theta)$ and advantage $\widehat{A}_{i,t}$ of token $y_{i,t}$ are:

$$r_{i,t}(\theta) = \frac{\pi_\theta(y_{i,t}|x, y_{i,<t})}{\pi_{\theta_{\text{old}}}(y_{i,t}|x, y_{i,<t})}, \quad \widehat{A}_{i,t} = \widehat{A}_i = \frac{r(x, y_i) - \text{mean}\left(\{r(x, y_i)\}_{i=1}^{G}\right)}{\text{std}\left(\{r(x, y_i)\}_{i=1}^{G}\right)}, \tag{2}$$

respectively, where all the tokens in $y_i$ share the same advantage as $\widehat{A}_i$.

A concise overview of PPO and GRPO is provided in Appendix A.1. Across all experiments, we adopt DAPO for fair comparison. In principle, any RLVR method can be applied to our TFPI stage.

# 3 METHODOLOGY

## 3.1 THINKING-FREE MODE ENABLES MORE EFFICIENT REASONING

To generate a response $y \sim \pi_\theta(\cdot \mid x)$ for a query $x$ using an SFT-distilled LRM $\pi_\theta$, the query is typically formatted with a chat template. Template 1 illustrates the template adopted by the Qwen model family (Yang et al., 2025a). We define *ThinkingFree* as an operator that transforms an input query $x$ into a modified query $x' = ThinkingFree(x)$, in which the thinking content is explicitly omitted. Under this transformation, response generation follows $y \sim \pi_\theta(\cdot \mid x')$. This mechanism provides explicit control over whether reasoning content is present or absent in the generated output (see Template 2). Additional examples using other chat templates are provided in Appendix A.2. Hereinafter, we use $x$ to denote a query formatted with the thinking template (e.g., Template 1), and $x'$ or *ThinkingFree*$(x)$ to denote the corresponding thinking-free version (e.g., Template 2).

> **Template 1 (Thinking Mode)** $<|im\_start|>system\backslash nPlease\ reason\ step\ by\ step,\ and\ put\ your$ *final answer within* $\backslash\backslash boxed\{\}.<|im\_end|>\backslash n<|im\_start|>user\backslash n\{$**question (x)**$\}$ $<|im\_end|>\backslash n<|im\_start|>assistant\backslash n$
>
> **Template 2 (Thinking-Free Mode)** $<|im\_start|>system\backslash nPlease\ reason\ step\ by\ step,\ and\ put$ *your final answer within* $\backslash\backslash boxed\{\}.<|im\_end|>\backslash n<|im\_start|>user\backslash n\{$**question (x)**$\}$ $<|im\_end|>\backslash n<|im\_start|>assistant\backslash n<think>\backslash n\backslash n</think>$

**During inference, converting $x$ into its thinking-free version $x'$ can substantially reduce token consumption.** To assess how this transformation affects the reasoning capability of SFT-distilled LRMs, we evaluate `DeepSeek-Distilled-Qwen-1.5B` (abbreviated as `DS-1.5B`) and `Qwen3-4B` on the AIME25. Detailed experimental setup is delayed to Appendix B.1. As shown in Figure 2 (Left), applying the *ThinkingFree* reduces the number of output tokens by more than 70% for both models. It is worth noting that `Qwen3-4B` is a fast–slow fusion model, whereas `DS-1.5B` is an SFT-distilled long-CoT model; nevertheless, both exhibit the same trend.

## 3.2 THINKING-FREE TRAINING IS BENEFICIAL TO SLOW-THINKING

To train an SFT-distilled LRM, the training response length should not be too short (Setlur et al., 2025), as this is detrimental to testing performance (An et al., 2025). As evidenced by the dotted line in Figure 2 (Right), training `Qwen-3-4B` with a 4K response length using DAPO indeed leads to a substantial drop in performance on AIME25. Given that using the *ThinkingFree* variant for inference can significantly reduce token consumption, we pose an audacious question: can we apply the *ThinkingFree* operation to all input queries during the rollout stage of RLVR? Moreover, will this approach be beneficial to preserving the original slow-thinking capability of the trained LLM?

Surprisingly, **RL trained with *ThinkingFree* rollout can slightly improve accuracy and reduce token consumption when evaluated in thinking mode, even with very short training context**

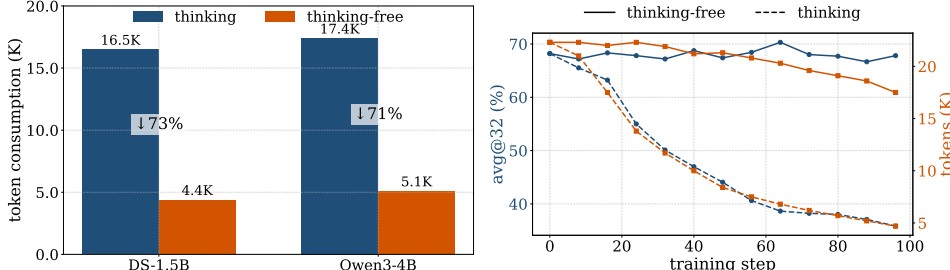

Figure 2: Results of the meta-experiment on the *ThinkingFree* operation. **Left:** Average output tokens in thinking mode and *ThinkingFree* mode on AIME25. **Right:** Evolution of `avg@32` and average output tokens on AIME24 with thinking-mode evaluation over training steps under 4K training response length (both training in thinking mode and *ThinkingFree* mode).

**lengths.** Figure 2 (Right) illustrates the training dynamics of `Qwen-3-4B` trained with a 4K response length under the *ThinkingFree* transformation of queries (detailed settings are in Appendix B.2). Even with a 4K output length, *ThinkingFree* RL increases the accuracy on AIME25 in thinking mode by approximately 2%, while reducing output tokens by around 20%. In contrast, standard RLVR with a 4K length reduces `avg@32` by more than 40%. These results suggest that applying *ThinkingFree* during rollout can yield steady improvements with minimal training compute.

### 3.3 THINKING FREE POLICY INITIALIZATION

Hu et al. (2025) introduce a pre-pretraining stage using formal language to accelerate convergence of pretraining, while Wang et al. (2025b) propose a mid-training stage between pretraining and RL-zero to facilitate RLVR. Inspired by these works, we ask: *can a dedicated stage for SFT-distilled LRMs improve the efficiency and effectiveness of standard RLVR scaling? ThinkingFree* RL, which enhances slow-thinking within short training context windows, requires substantially less compute than standard RLVR. Initializing RLVR with a *ThinkingFree* RL policy could therefore reduce rollout tokens while achieving stronger downstream performance. We term this step as **T**hinking **F**ree **P**olicy **I**nitialization (**TFPI**), a stage preceding standard RLVR for SFT-distilled LRMs that aims to lower rollout costs, raise the ceiling of reasoning ability, and accelerate convergence.

Consider the RLVR objective $\mathcal{J}_{\text{RLVR}}(\theta) = \mathbb{E}_{x \sim \mathcal{D}} \left[ \mathcal{J}_{\text{RLVR}}(\theta, x) \right]$, where $\mathcal{J}_{\text{RLVR}}(\theta, x)$ denotes the per-example objective of any RLVR algorithm (e.g., DAPO in eq. (1) or GRPO in eq. (5)). For the TFPI stage, we use a modified objective: $\mathcal{J}_{\text{TFPI}}(\theta) = \mathbb{E}_{x \sim \mathcal{D}} \left[ \mathcal{J}_{\text{RLVR}}(\theta, x') \right]$, where $x' = ThinkingFree(x)$. In the rollout stage of TFPI, $G$ responses are generated conditioned on $x'$: $\{y_i\}_{i=1}^G \sim \pi_{\theta_{\text{old}}}(\cdot \mid x')$. The importance ratios and advantages (eq. (2)) are then adapted as:

$$r_{i,t}(\theta) = \frac{\pi_\theta(y_{i,t} \mid x', y_{i,<t})}{\pi_{\theta_{\text{old}}}(y_{i,t} \mid x', y_{i,<t})}, \quad \widehat{A}_{i,t} = \widehat{A}_i = \frac{r(x', y_i) - \text{mean}\big(\{r(x', y_j)\}_{j=1}^G\big)}{\text{std}\big(\{r(x', y_j)\}_{j=1}^G\big)}, \quad (3)$$

where Template 1 and Template 2 showcase the correspondence between $x$ and $x'$ and $r(x', y) = r(x, y)$ since the *ThinkingFree* operator does not alter the ground-truth answer of the original problem. In our experiments, we instantiate RLVR with DAPO, i.e., $\mathcal{J}_{\text{RLVR}}(\theta) = \mathcal{J}_{\text{DAPO}}(\theta)$ (see eq. (1)). Note that Heuristics for RLVR (e.g., multi-stage RL, data curricula) can be directly applied to TFPI.

## 4 EXPERIMENTS

### 4.1 EXPERIMENTAL SETTINGS

In this section, we provide a brief overview of the key experimental setup, including training procedures, baselines, and evaluation details. Additional information can be found in Appendix C.

**Training Details**. We build on the VeRL codebase (Sheng et al., 2024) with RLVR, following the DAPO recipe (Yu et al., 2025), which enables dynamic sampling and clipping higher. All methods use the same hyperparameters (batch size 256, learning rate $1 \times 10^{-6}$, no warm-up) and rollout settings (temperature 1, `topp` 1, `topk` $-1$, 8 rollouts/problem). Training is conducted on `DS-1.5B`,

Table 1: Results of TFPI vs. direct RL across different benchmarks. "`Avg@k`" denotes the average accuracy (%) over $k$ random generations (i.e., `pass@1`). All models are evaluated in thinking mode. The total training compute for the 3 stages of TFPI equals that of "Direct RL" for fair comparison. Darker colors in the cell background denote better results within each model group.

| Models | Mathematics | | | Multi-Task | Code | Instruction | Overall |
|---|---|---|---|---|---|---|---|
| | **AIME 24**
Avg@32 | **AIME 25**
Avg@32 | **Beyond AIME**
Avg@8 | **GPQA**
Avg@8 | **LiveCode**
Avg@8 | **IFEval**
Avg@4 | **Overall**
Avg. |
| **DeepSeek-Distill-Qwen-1.5B** | | | | | | | |
| Initial Model | 29.6 | 23.0 | 8.7 | 16.3 | 17.7 | 36.6 | 22.0 |
| - Direct RL | 33.9 | 27.1 | 11.9 | 19.2 | 18.2 | 41.8 | 25.3 |
| - TFPI stage 1 | 32.1 | 26.9 | 13.9 | 29.3 | 18.4 | 39.5 | 26.7 |
| - TFPI stage 2 | 36.8 | 28.4 | 14.5 | 27.8 | 20.5 | 42.3 | 28.4 |
| - TFPI stage 3 | 40.1 | 30.8 | 13.8 | 29.6 | 19.9 | 40.8 | 29.2 |
| **Qwen3-4B** | | | | | | | |
| Initial Model | 73.6 | 68.3 | 43.4 | 56.8 | 54.9 | 64.9 | 60.3 |
| Direct RL | 75.7 | 67.0 | 43.6 | 56.3 | 52.5 | 66.0 | 60.2 |
| TFPI stage 1 | 75.2 | 67.8 | 42.4 | 57.9 | 55.3 | 66.0 | 60.8 |
| TFPI stage 2 | 76.0 | 68.2 | 44.7 | 57.8 | 54.8 | 64.8 | 61.0 |
| TFPI stage 3 | 79.9 | 70.6 | 46.7 | 58.5 | 57.0 | 70.2 | 63.8 |
| **DeepSeek-Distill-Qwen-7B** | | | | | | | |
| Initial Model | 56.3 | 40.0 | 25.0 | 36.9 | 39.5 | 55.3 | 42.2 |
| - Direct RL | 57.9 | 40.4 | 26.6 | 38.0 | 38.3 | 56.7 | 43.0 |
| - TFPI stage 1 | 59.4 | 40.4 | 29.2 | 49.0 | 39.6 | 56.2 | 45.6 |
| - TFPI stage 2 | 61.8 | 43.9 | 31.5 | 48.0 | 42.0 | 57.1 | 47.4 |
| - TFPI stage 3 | 62.0 | 44.6 | 31.1 | 46.8 | 42.1 | 60.2 | 47.8 |

`Qwen3-4B`, and `DS-7B` using Polaris-53K (An et al., 2025). Direct RLVR uses a maximum output length of 16K for `DS-1.5B`/`DS-7B` and 32K for `Qwen3-4B`, while TFPI adopts multi-stage training: 2K→4K→8K for `DS-1.5B`/`DS-7B` and 4K→8K→16K for `Qwen3-4B`.

**Baselines**. We compare TFPI with direct RLVR training from an SFT-distilled LRM ("Direct RL") under matched total training compute (Table 1). To assess TFPI as a pre-RLVR stage, we run "TFPI + RL" with similar compute and compare against "Direct RL" (Table 2), also including competitive LRMs of the same size, such as Polaris (An et al., 2025), DeepScaleR (Luo et al., 2025b), Skywork-OR1 (He et al., 2025), and so on. For efficiency analysis, we compare with several RL-based efficient reasoning baselines (Table 3) under the same evaluation for fair comparison, including TLMRE (Arora & Zanette, 2025), AdaptThink (Zhang et al., 2025b), AutoThink (Tu et al., 2025), Laser (Liu et al., 2025a), L1Max (Aggarwal & Welleck, 2025), and ThinkLess (Fang et al., 2025).

**Evaluation Details**. Our evaluation benchmarks cover: ❶ **Math Reasoning**: AIME24/25 and BeyondAIME (ByteDance-Seed, 2025). ❷ **Multi-Task Reasoning**: GPQA-Diamond (Rein et al., 2024). ❸ **Code Generation**: LiveCodeBench (Jain et al., 2024). ❹ **Instruction Following**: IFEval (Zhou et al., 2023). Following Guo et al. (2025), we generate multiple outputs (ranging from 4 to 32 depending on the size of the test set) per problem and report `pass@1` accuracy. Note that ❶ is in-domain evaluation, and ❷ ❸ ❹ are out-of-domain evaluation. For IFEval, we report the *strict prompt* accuracy. All evaluation scripts are adapted from the DeepscaleR codebase (Luo et al., 2025b). Following the officially recommended decoding parameters*, we set the temperature to 0.6 and top-p to 0.95 for the thinking mode, and 0.7, 0.8 for the thinking-free mode, respectively. The maximum output length is 32k for all configurations, except for the `Qwen3-4B` thinking mode, where 48k is used to reduce the clipping ratio. Further details are provided in Appendix C.3, and the effects of changing decoding parameters are presented in Appendix F.4.

## 4.2 TFPI ENHANCES THE SLOW-THINKING OF DISTILLED REASONING MODELS

To evaluate the impact of TFPI on the slow-thinking capabilities, we present the results of TFPI versus "Direct RL" under the same training compute in Table 1. From the table, we have:

❶ **TFPI substantially enhances the slow-thinking capabilities of distilled LRMs even when trained with a small response length.** For example, on `DS-1.5B`, TFPI Stage 1 raises the overall

---

*See Qwen3-4B Hugging Face page and DS-1.5B Hugging Face page.

Table 2: Results (%) of RL after TFPI ("TFPI+RL") vs. "Direct RL" across different benchmarks. "Avg@k" denotes the average accuracy (%) over $k$ random generations (i.e., `pass@1`). For LRMs marked with "*", results are taken from the corresponding reports (see Appendix C.4); results of 4B models are from our own runs with 48K response length. All models are evaluated in thinking mode. The total training compute for "TFPI+RL" is matched to that of "Direct RL" for fair comparison. Darker colors in the cell background denote better results.

| Models | Mathematics | | | Multi-Task | Code | Instruction | Overall |
|---|---|---|---|---|---|---|---|
| | AIME 24 Avg@32 | AIME 25 Avg@32 | Beyond AIME Avg@8 | GPQA Avg@8 | LiveCode Avg@8 | IFEval Avg@4 | Overall Avg. |
| DeepSeek R1* | 79.8 | 65.0 | 42.4 | 71.5 | 64.3 | 86.1 | 68.2 |
| Seed-1.5-Thinking* | 86.7 | 74.0 | 48.0 | 77.3 | 64.9 | 87.4 | 73.0 |
| Claude4 Opus Thinking* | - | 75.5 | - | 79.6 | 48.9 | 89.7 | - |
| Qwen3-235B-Thinking* | 85.7 | 81.5 | - | 71.1 | 55.7 | 83.4 | - |
| Qwen3-4B Direct RL | 78.8 | 71.5 | 46.4 | 56.2 | 54.3 | 65.1 | 62.0 |
| Qwen3-4B TFPI stage 3 | 79.9 | 70.6 | 46.7 | 58.5 | 57.0 | 70.2 | 63.8 |
| Qwen3-4B TFPI + RL | 80.8 | 76.0 | 49.5 | 61.1 | 55.7 | 71.3 | 65.7 |
| Qwen3-4B-2507-Thinking | 84.7 | 79.2 | 51.5 | 66.4 | 62.4 | 68.0 | 68.7 |
| - TFPI only | 89.0 | 81.2 | 51.3 | 70.1 | 65.5 | 66.7 | 70.6 |

average accuracy from 22.0% to 26.7% (+4.7%) despite being restricted to a 2K training response length. Results of `DS-1.5B` continue to improve through stages 1 to 3 on AIME25, accuracy increases from 23.0% (initial model) to 26.9% after Stage 1, 28.4% after Stage 2, and 30.8% after Stage 3, representing a total gain of +7.8%. Similar improvements are observed for `Qwen3-4B` and `DS-7B`. These findings indicate that TFPI enables effective training under low-cost settings (short context windows), and that combining this with multi-stage RL yields substantial accuracy gains.

❷ **Compared with "Direct RL", TFPI delivers faster and larger accuracy improvements under the same training cost.** TFPI outperforms "Direct RL" in nearly all configurations. For example, `Qwen3-4B` with TFPI attains 63.8% overall accuracy versus 60.2% for "Direct RL" (+3.6%), while `DS-7B` improves by +4.8% (47.8% vs. 43.0%). Given that the equal training compute, these results imply that TFPI achieves convergence more efficiently than conventional long-CoT RL training.

❸ **Improvements of TFPI indicate some degree of transfer beyond mathematics.** Although TFPI is trained solely on Polaris-53K (math-specific data), it demonstrates great out-of-domain improvements as well. For example, on `DeepSeek-Distill-Qwen-1.5B`, GPQA accuracy increases from 16.3% to 29.6%, LiveCodeBench from 17.7% to 19.9%, and IFEval from 36.6% to 40.8% after Stage 3. Notably, **improvements on mathematical benchmarks are often consistent across successive training stages, whereas other domains sometimes exhibit fluctuations** (e.g., GPQA for `DS-7B` and LiveCodeBench for `DS-1.5B`). This suggests that **incorporating more diverse training data spanning multiple domains could be helpful for TFPI**.

### 4.3 TFPI as a Foundation for RLVR to Achieve Higher Performance

Table 2 shows the representative results between "TFPI + RL" with "Direct RL" under the same training cost (full results in Table 9 in Appendix F.2). The results lead to the following observations:

❶ **TFPI can raise the upper bound of RL-trained performance.** RL training on a TFPI-trained model can still yield notable improvements, particularly in mathematics. For example, adding an RL stage after TFPI for `Qwen3-4B` boosts AIME25 accuracy from 70.6% to 76.0% (+5.4%). Similarly, Beyond AIME scores increase by more than 2%. Across different model scales, applying RL after TFPI consistently achieves higher accuracies than "Direct RL" under the same compute budget. For instance, on `Qwen3-4B`, the overall accuracy rises from 62.0% (Direct RL) to 65.7% (TFPI+RL).These results suggest that incorporating TFPI as an intermediate stage between long CoT distillation and standard RLVR can be beneficial for elevating final performance.

❷ **TFPI + RL is an effective and efficient strategy for training high-performing LRMs.** From Figure 1 (Left), the total compute cost of all three TFPI stages amounts to less than 20% of standard RL training with 32K token sequences. From Table 2, for `DS-7B`, TFPI+RL achieves approximately the same overall performance as `Polaris-7B-Preview`, despite using a maximum response length of 16K, whereas `Polaris-7B-Preview` follows a 16K → 24K → 32K progression during RL. Similarly, `Polaris-4B-Preview` employs a 40K → 48K → 52K length

Table 3: Comparison of the Thinking-Free inference mode of TFPI with efficient reasoning baselines across various reasoning tasks. "`Avg@k`" denotes the average accuracy (in %) over k generations (i.e., `pass@1`), and "`Toks`" indicates the average output length in thousands of tokens (K). Models with "*" are trained from `DeepScaleR-1.5B`, while the remaining are from `DS-1.5B`. Darker cell background colors indicate better results.

| Models | AIME 24 | | AIME 25 | | Beyond AIME | | GPQA | | Overall | |
|---|---|---|---|---|---|---|---|---|---|---|
| | avg@32 | Toks | avg@32 | Toks | avg@8 | Toks | avg@8 | Toks | Avg. | Toks |
| 1) TLMRE-DS-1.5B ($\alpha = 0.1$) | 27.6 | 12.9 | 24.8 | 12.5 | 9.2 | 11.9 | 14.8 | 7.5 | 19.1 | 11.2 |
| 2) AdaptThink-1.5B ($\delta = 0.05$) | 28.1 | 8.0 | 22.6 | 7.9 | 10.0 | 4.8 | 14.8 | 5.1 | 18.9 | 6.5 |
| 3) AutoThink-DS-1.5B-Stage3 | 30.3 | 10.2 | 25.2 | 9.1 | 9.4 | 8.7 | 17.4 | 7.0 | 20.6 | 8.7 |
| 4) Laser-DE-L4096-1.5B | 30.3 | 8.3 | 24.9 | 7.4 | 9.7 | 7.4 | 21.1 | 4.7 | 21.5 | 6.9 |
| 5) AutoThink-Stage3* | 38.9 | 8.7 | 28.9 | 7.7 | 11.6 | 7.8 | 27.3 | 5.4 | 26.7 | 7.4 |
| 6) L1-1.5B-Max* | 27.2 | 3.2 | 26.3 | 2.9 | 9.1 | 3.1 | 32.4 | 2.3 | 23.8 | 2.9 |
| 7) Thinkless-1.5B-RL* | 28.4 | 11.3 | 24.1 | 11.1 | 8.1 | 11.7 | 20.3 | 12.7 | 20.2 | 11.7 |
| DS-1.5B (Thinking) | 29.6 | 16.7 | 23.0 | 16.5 | 8.7 | 14.4 | 16.3 | 9.8 | 19.4 | 14.3 |
| DS-1.5B (Thinking-Free) | 12.4 | 5.7 | 10.9 | 4.4 | 4.4 | 3.4 | 14.2 | 0.9 | 8.0 | 3.6 |
| - TFPI stage 1 | 21.9 | 1.6 | 15.3 | 1.4 | 8.7 | 1.3 | 32.9 | 0.8 | 19.7 | 1.3 |
| - TFPI stage 2 | 31.5 | 3.4 | 24.2 | 3.1 | 10.1 | 2.9 | 35.3 | 1.6 | 25.3 | 2.7 |
| - TFPI stage 3 | 37.5 | 5.3 | 28.4 | 5.0 | 12.4 | 4.9 | 35.6 | 2.6 | 28.5 | 4.4 |

schedule and consumes approximately 8K H800 GPU hours, while our TFPI+RL requires only about 1.5K H800 GPU hours (see Appendix C.1) and achieves superior performance under 48K testing length. Even without a subsequent RL stage, TFPI alone allows `DS-1.5B` to outperform `DeepScaleR`, which uses a maximum training length of 24K. Using only TFPI (4K→8K→16K), `Qwen3-4B-2507` achieves 89% accuracy on AIME24. Remarkably, this 4B model outperforms `Qwen3-235B-Thinking` in math reasoning and code generation. These findings suggest that TFPI can serve both as a strong standalone training approach and as an efficient foundation for subsequent RL, producing competitive LRMs with significantly reduced compute requirements.

## 4.4 TFPI Improves the Token Efficiency of Distilled Reasoning Models

We compare the thinking-free inference with other RL-based efficient reasoning baselines in 1.5B size in Table 3 (see also Table 10 in Appendix F.2). We draw the following conclusions:

❶ **Both accuracy and token efficiency steadily improve after stage 3 of TFPI.** For `DS-1.5B`, thinking-free accuracy on AIME24 increases from 29.6% (initial model) to 37.5% (TFPI stage 3), while the average output length remains substantially shorter than that of the original thinking mode (5.3K vs. 16.7K tokens). A similar trend is observed for `Qwen3-4B`, where accuracy improves from 26.9% to 75.1% across stages, with output lengths still far below those of the original thinking model. These results demonstrate that TFPI naturally produces more token-efficient LRMs, offering an alternative pathway to train models that deliver both high accuracy and reduced output length.

❷ **Compared with other RL-based token control methods, TFPI achieves the best performance–efficiency trade-off without specialized reward or training designs.** In `DS-1.5B`, both TFPI stage 2 and stage 3 outperform almost all baselines in terms of overall accuracy while maintaining competitive or lower token usage. We visualize the overall accuracy–token usage trade-off in Figure 1 (Right), where TFPI consistently lies on the Pareto frontier across different stages. This observation motivates a rethinking of existing "token-efficient reasoning RL designs" that rely heavily on specialized length reward shaping: training with TFPI offers an alternative paradigm in which a strong slow-thinking LRM can be obtained, and a more efficient variant can be realized simply by switching to the thinking-free mode without any additional length-control mechanisms.

## 5 Analysis

### 5.1 Why TFPI Leads to Better Thinking-Mode Inference?

In this section, we delve deeper into why TFPI, which leverages void thinking content in Template 2, can generalize to enhance reasoning in Template 1. We analyze `DS-1.5B` from two perspectives:

❶ **Behavioural level:** The learned *verification* behaviours after `</think>` during TFPI can generalize to the slow-thinking verification occurring within the `<think>` and `</think>`. The blue lines in Figure 3 show the ratio of verification steps (i.e., the number of verification steps divided by

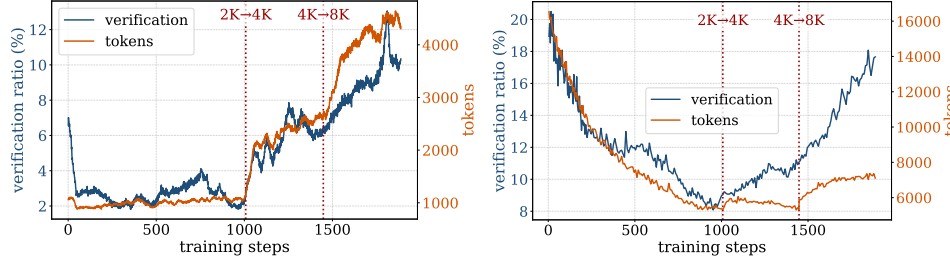

Figure 3: Behaviour-Level Analysis of `DS-1.5B` over the TFPI Training Course. The ratio of verification steps and the average output tokens over training steps on the training set in thinking-free mode (**Left**) and on AIME25 in thinking mode (**Right**) in 3 stages of TFPI.

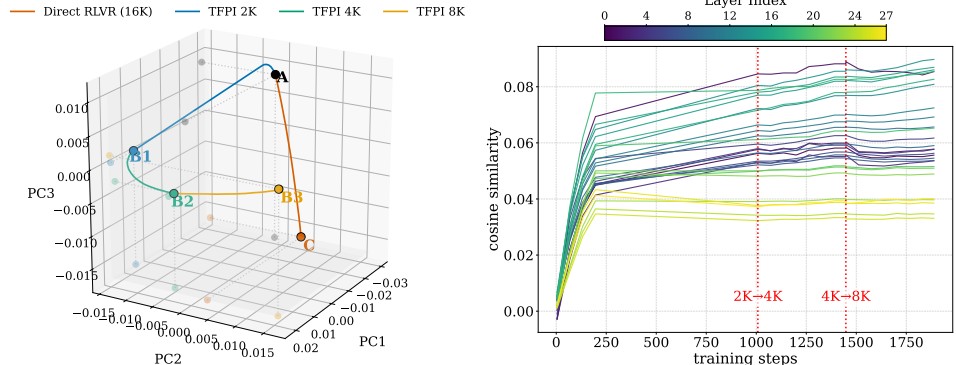

Figure 4: Parameter-Level Analysis. **Left:** PCA projection of model parameters from `DS-1.5B` to final checkpoints. TFPI (blue) starts at **A**, passes through intermediate points (**B1**, **B2**, **B3**), and ultimately converges near the Direct RL final checkpoint (**C**). **Right:** Cosine similarity between parameter updates of TFPI-trained checkpoints and (C-A) across layers during training.

the total number of steps; see Appendix D for details) for the training set (thinking-free mode) and AIME 25 dataset (thinking mode). We observe that the verification ratio exhibits a similar trend: a rapid drop in Stage 1, followed by steady growth in Stage 2, and a sharp increase in Stage 3. Notably, the sharp decline in Stage 1 resembles an information compression process. In Stages 2 and 3, the model begins to explore more extensively (Figure 3 Right), which may explain why TFPI achieves superior performance. As verification is believed to be vital for slow-thinking reasoning (Setlur et al., 2025), the observed generalization of verification behavior suggests a transfer from thinking-free training to inference in thinking mode.

❷ **Parameter level:** TFPI explores the parameter space more extensively at a faster pace, with its parameter update directions progressively aligning with those of "Direct RL." As shown in Figure 4 (Left), the PCA visualization of the initial model, TFPI-trained checkpoints, and Direct RL checkpoints exhibits distinct trajectories in parameter space. The TFPI begins from the initial model (**A**), moves through intermediate points (**B1**), (**B2**), and (**B3**), and ultimately converges to a region near the Direct RL final checkpoint (**C**). This indicates that TFPI traverses a larger and more diverse region of parameter space before reaching a point close to the RL-trained model. Such broad exploration may help explain why TFPI can lead to better LRMs. Furthermore, Figure 4 (Right) shows that the cosine similarity of parameter updates between the TFPI-trained checkpoints and the Direct RL final checkpoint steadily increases across nearly all layers throughout training. This suggests that during TFPI training, the parameter updates share similarities with those in standard long-CoT training. To make our analysis more robust, additional results are given in Appendix F.1.

## 5.2 ABOUT REASONING PATTERN AND ROLLOUT SPEED

**TFPI Preserves the Reasoning Pattern of Thinking Mode.** In thinking mode (Template 1), a response $y$ comprises a long thinking section and a concise answer $y^{\text{ans}}$. In *ThinkingFree* mode (Template 2), the thinking section is omitted, and $y^{\text{ans}}$ contains a shorter reasoning path. While standard RL tends to lengthen the thinking part, TFPI increases the length of $y^{\text{ans}}$ due to the absence of explicit thinking. As shown in Figure 5 (Left), for `DS-1.5B` trained with TFPI and evaluated in

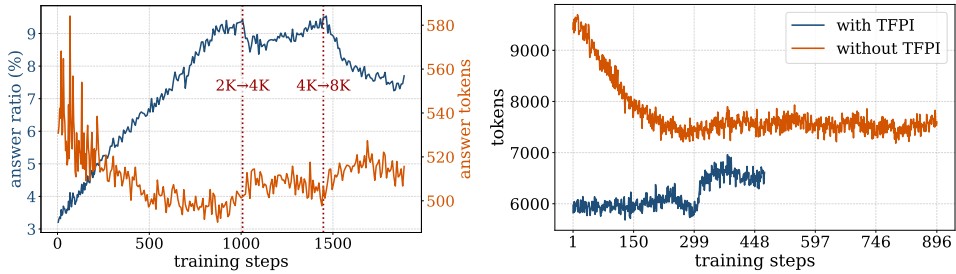

Figure 5: **Left:** For TFPI with `DS-1.5B` on AIME25 in thinking mode, showing the average number of answer tokens (excluding the thinking part) and the ratio of answer length to total response length over training steps. **Right:** For long CoT RL with `DS-1.5B`, showing the average number of tokens during rollout on the training set over training steps, with and without the TFPI stage.

Table 4: Additional results of TFPI. All models are evaluated in thinking mode.

| Models | AIME 24 | AIME 25 | Beyond AIME | GPQA | LiveCode | IFEval | Overall |
|---|---|---|---|---|---|---|---|
| **Stage Schedule Sensitivity, Qwen3-4B** | | | | | | | |
| 4K → 8K → 16K | **79.9** | 70.6 | **46.7** | 58.5 | **57.0** | **70.2** | **63.8** |
| 8K → 16K | 78.0 | 70.2 | **46.7** | **59.5** | 55.8 | 64.5 | 62.4 |
| 16K only | 77.1 | **71.5** | 45.8 | 56.4 | 54.4 | 66.0 | 61.9 |
| **Multi-Stage Direct RL, Qwen3-4B** | | | | | | | |
| TFPI Stage 3 | **79.9** | **70.6** | **46.7** | **58.5** | **57.0** | **70.2** | **63.8** |
| Multi-Stage Direct RL | 64.9 | 48.6 | 32.5 | 55.2 | 51.0 | 65.2 | 52.9 |
| **Qwen3-14B** | | | | | | | |
| Initial Model | 81.7 | 73.2 | 50.2 | 65.0 | 63.5 | 67.3 | 66.8 |
| - Direct RL | 81.7 | 74.5 | **50.8** | 64.8 | 61.8 | **68.2** | 66.9 |
| - TFPI stage 3 | **83.2** | **76.0** | 50.5 | **65.6** | **63.8** | 67.8 | **67.8** |
| **Qwen3-14B Tokens (K)** | | | | | | | |
| Initial Model | 16.8 | 20.8 | 22.8 | 9.4 | 16.0 | - | 17.1 |
| - Direct RL | 17.9 | 22.1 | 24.5 | 10.4 | 17.4 | - | 18.5 |
| - TFPI stage 3 | **13.8** | **17.3** | **18.3** | **7.3** | **13.8** | - | **14.1** |

thinking mode, $|y^{\mathrm{ans}}|$ remains stable at 500–580 tokens, whereas $|y^{\mathrm{ans}}|/|y|$ rises as the total length $|y|$ decreases. This indicates that TFPI preserves the core reasoning pattern of slow-thinking rather than drifting toward an excessively extended "slow-slow thinking" behavior.

**TFPI Speeds Up Rollout for Long-CoT RL Training**. Another advantage of TFPI is its ability to speed up the rollout stage in standard long-CoT RL training. As shown in Figure 5 (Right), when directly performing RL from `DS-1.5B`, the average output tokens during rollout on the training set start at over 9K, decrease to around 7.5K within the first 300 steps, and then fluctuate around 7.5K in the later stages. In contrast, when RL is performed after the TFPI phase, the average output tokens start at only 6K and the maximum length is below 7K tokens.

## 5.3 Scaling Up Model Size

To assess the effect of TFPI on larger reasoning models, we conducted experiments with `Qwen3-14B` under the same settings as `Qwen3-4B`. As shown in Table 4, TFPI still outperforms Direct RL under the same training compute. Notably, Direct RL yields results only comparable to those of the initial model (66.9% vs. 66.8% on average). We hypothesize that Polaris-53K is, in some sense, too easy for `Qwen3-14B` (with over 78% rollout accuracy), leaving only approximately 30% effective data after dynamic sampling (excluding 0/8 and 8/8 prompts). Nonetheless, TFPI remains effective in this setting. Additionally, TFPI reduces the average number of output tokens on the test set by 23.8% even in thinking mode (14.1K vs. 18.5K). These results highlight the effectiveness of TFPI for larger model sizes, and we believe that with more challenging, higher-quality data, TFPI can further realize its scalability potential.

## 5.4 ABLATION STUDY

**Stage Schedule Sensitivity**. To assess sensitivity to the stage schedule, we train `Qwen3-4B` under alternative schedules (8K→16K and 16K-only) with approximately matched compute budgets. As shown in Table 4, all schedules outperform the Direct RL baseline and achieve comparable performance on math benchmarks, indicating that TFPI is robust to the choice of schedule. Nevertheless, $4K \rightarrow 8K \rightarrow 16K$ yields the best overall results (63.8% vs. 62.4% and 61.9%). We hypothesize that this stems from an implicit data curriculum induced by dynamic sampling: beginning at 4K improves training efficiency and emphasizes easier problems early on.

**Multi-Stage Direct RL**. As discussed in Section 3.2 and Figure 2 (right), applying Direct RL with a 4K response length causes a substantial performance drop on AIME25 for `Qwen3-4B`. In contrast, TFPI trains stably even at 4K. This indicates that, even in a multi-stage setting, the initial stage length should not be too short. To rule out the potential artifact introduced by multi-stage training, we run Direct RL with the identical multi-stage schedule on `Qwen3-4B`. As shown in Table 4, Direct RL still incurs a substantial performance decrease under the $4K \rightarrow 8K \rightarrow 16K$ schedule. Additional details are provided in Appendix F.7.

## 6 RELATED WORK

**LRMs & Efficiency RLVR**. RLVR has enabled the development of numerous high-performing large reasoning models (LRMs) (Yang et al., 2025a; Zeng et al., 2025a; An et al., 2025), inspiring research on model behaviors (Liu et al., 2025b; Wang et al., 2025a), novel algorithms (Liu et al., 2025b; Zheng et al., 2025), multimodal extensions (Meng et al., 2025; Xiao et al., 2025; Wang et al., 2025c), tool integration (Jin et al., 2025; Feng et al., 2025; Li et al., 2025a; Xue et al., 2025; Team et al., 2025a), and other related directions (Xu et al., 2025a; Zhang et al., 2025a; Chen et al., 2025c; Zhang et al., 2025c). RLVR can be applied directly to base LLMs ("RL zero") (Zeng et al., 2025b; Guo et al., 2025) or initialized from SFT-distilled long-CoT models, the latter typically yielding stronger results (Luo et al., 2025b; An et al., 2025). A major challenge for RLVR is the cost of training with long contexts, as longer outputs are often necessary for harder tasks (Shrivastava et al., 2025; Zeng et al., 2025a), consuming high computational resources. Multi-stage RLVR mitigates this by starting with shorter contexts and gradually extending them (Luo et al., 2025b; An et al., 2025; He et al., 2025), while algorithmic approaches modify GRPO to reduce length bias (Yu et al., 2025; Liu et al., 2025b; Wang et al., 2025a). Orthogonal to these strategies, we introduce TFPI as a lightweight stage before RLVR, improving efficiency and strengthening the slow-thinking mode at inference with minimal training cost, thus facilitating more effective subsequent RLVR.

**Efficient Reasoning**. To address the issue of overthinking (Chen et al., 2024; Sui et al., 2025), considerable efforts have been made, including prompt-based methods (Muennighoff et al., 2025; Yang et al., 2025c; Fu et al., 2025a; Chen et al., 2025a; Fu et al., 2025b), SFT-based approaches (Kang et al., 2025; Ma et al., 2025; Munkhbat et al., 2025; Luo et al., 2025a), and RL-related designs. RL-related approaches can be further categorized into length-based reward shaping (Team et al., 2025b; Aggarwal & Welleck, 2025; Arora & Zanette, 2025; Liu et al., 2025a), integration of fast and slow thinking (Fang et al., 2025; Zhang et al., 2025b; Lou et al., 2025; Tu et al., 2025; Jiang et al., 2025; Zhang et al., 2025d), and thinking budget control (Li et al., 2025b; Hammoud et al., 2025; Wen et al., 2025). These methods primarily trade accuracy for efficiency and rely on specialized reward functions or training strategies to encourage more efficient reasoning. In contrast, our proposed TFPI naturally yields even more efficient LRMs without specialized rewards or training designs.

## 7 CONCLUSION

After recognizing the benefits of *ThinkingFree* for both inference and the training of distilled reasoning models, we introduce TFPI, a cost-efficient intermediate stage between long-CoT distillation and standard RL training. As a strong initialization point, TFPI accelerates RL convergence, enhances attainable performance, and promotes more token-efficient reasoning without complex reward shaping or elaborate training pipelines. We further explain the factors behind TFPI's success from both the behavioral and parameter levels. Overall, TFPI provides a complementary path for building "token-efficient" LRMs, offering an effective and efficient alternative to current RL paradigms.

ACKNOWLEDGEMENTS

This work was partially supported by an Area of Excellence project (AoE/E-601/24-N), a Theme-based Research Project (T32-615/24-R), and the Innovation and Technology Commission (ITCPD/179) from the Research Grants Council of the Hong Kong Special Administrative Region, China. We would like to thank all reviewers for their helpful suggestions in improving this paper.

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

# A  BACKGROUND AND PRELIMINARY

## A.1  RLVR ALGORITHMS

**Proximal Policy Optimization (PPO)**. PPO (Schulman et al., 2017) constrains the policy update within a proximal region of the old policy $\pi_{\theta_{old}}$ through the clipping mechanism. Specifically, PPO employs the following objective (we omit the KL regularization term hereinafter for brevity):

$$\mathcal{J}_{\text{PPO}}(\theta) = \mathbb{E}_{x \sim \mathcal{D}, \, y \sim \pi_{\theta_{old}}(\cdot|x)} \left[ \frac{1}{|y|} \sum_{t=1}^{|y|} \min \left( r_t(\theta) \widehat{A}_t, \, \text{clip}\left( r_t(\theta), 1 - \varepsilon, 1 + \varepsilon \right) \widehat{A}_t \right) \right], \tag{4}$$

with the importance ratio of the token $y_t$ is defined as $r_t(\theta) = \frac{\pi_\theta(y_t|x, y_{<t})}{\pi_{\theta_{old}}(y_t|x, y_{<t})}$, the advantage $\widehat{A}_t$ of $y_t$ is estimated by a value model, and $\varepsilon$ is the clipping range of importance ratios.

**Group Relative Policy Optimization (GRPO)**. GRPO (Shao et al., 2024) bypasses the need for the value model by computing the relative advantage of each response within a group of responses to the same query. Specifically, GRPO optimizes $\mathcal{J}_{\text{GRPO}}(\theta) = \mathbb{E}_{x \sim \mathcal{D}} \left[ \mathcal{J}_{\text{GRPO}}(\theta, x) \right]$, where:

$$\mathcal{J}_{\text{GRPO}}(\theta, x) = \left[ \frac{1}{G} \sum_{i=1}^{G} \frac{1}{|y_i|} \sum_{t=1}^{|y_i|} \min \left( r_{i,t}(\theta) \widehat{A}_{i,t}, \, \text{clip}\left( r_{i,t}(\theta), 1 - \varepsilon, 1 + \varepsilon \right) \widehat{A}_{i,t} \right) \right]. \tag{5}$$

Here $\{y_i\}_{i=1}^G \sim \pi_{\theta_{old}}(\cdot|x)$, $G$ is the number of generated responses to each query $x$ (i.e., the group size), and $r_{i,t}(\theta)$, $\widehat{A}_{i,t}$ is computed as in eq. (2).

Numerous variants have been proposed to improve GRPO. For example, DAPO (Yu et al., 2025) introduces token-level normalization and dynamic sampling; Dr GRPO (Liu et al., 2025b) removes length bias to prevent incorrect responses from growing longer over time; and Wang et al. (2025a) train selectively on forking tokens (see also Section 6). Our TFPI is orthogonal to these RLVR algorithms. That is to say, any RLVR algorithm can be applied to our proposed TFPI stage. **To isolate the effect of RLVR algorithms, we employ DAPO as our RLVR algorithm in all experiments for fair comparison.**

## A.2  *ThinkingFree* OPERATION

In Section 3.1, we have shown chat templates for both the original query $x$ and its thinking-free version $x' = \textit{ThinkingFree}(x)$ for Qwen models. Here, we showcase one additional example under the DeepSeek (Guo et al., 2025) template as below.

> **Template 3 (Thinking Mode (DeepSeek))** *Please reason step by step, and put your final answer within* \\*boxed{}.<|User|>***{question (x)}***<|Assistant|>\n*
>
> **Template 4 (Thinking-Free Mode (DeepSeek))** *Please reason step by step, and put your final answer within* \\*boxed{}.<|User|>***{question (x)}***<|Assistant|>\n<think>\n\n</think>*

# B  META EXPERIMENTS

## B.1  TOKEN CONSUMPTION OF *ThinkingFree*

We provide the detailed experimental setup for the meta-experiment in Section 3.1 in this appendix.

For `DS-1.5B`, we use the decoding parameters suggested by Guo et al. (2025) in thinking mode: temperature = 0.6, top-$p$ = 0.95, and top-$k$ = −1. For `Qwen3-4B`, we adopt the recommended settings from Yang et al. (2025a) in thinking mode: temperature = 0.6, top-$p$ = 0.95, and top-$k$ = 20. For *ThinkingFree*, we set temperature = 0.7, top-$p$ = 0.8, and use both top-$k$ = −1 and top-$k$ = 20. The maximum output length is fixed at 32K tokens for both modes. For evaluation, we sample 32 generations per query on AIME 2025 and report the average number of output tokens. The parameters are given in Table 5 and the results are shown in Figure 2 (Left).

Table 5: Decoding Parameters of Meta-Experiment in Section 3.1

| | Thinking Mode | Thinking-Free Mode |
|---|---|---|
| `DS-1.5B` | $T = 0.6$, top-$p = 0.95$, and top-$k = -1$ | $T = 0.7$, top-$p = 0.8$, and top-$k = 20$ |
| `Qwen3-4B` | $T = 0.6$, top-$p = 0.95$, and top-$k = 20$ | $T = 0.7$, top-$p = 0.8$, and top-$k = 20$ |

### B.2 Detailed Setup of *ThinkingFree* Training

We provide the detailed experimental setup for the meta-experiment discussed in Section 3.2 in this appendix.

For training, we use the DAPO (Yu et al., 2025) algorithm, with configurations identical to those in Appendix C.1, except that the maximum output length is set to 4K. For evaluation, we set the maximum output length to 48K and perform testing in thinking mode as described in Template 1. All other evaluation parameters follow Appendix C.3. Note that the initial avg@32 value in Figure 2 (Right) is higher than the value reported in (Yang et al., 2025a) (68.2 vs. 65.6), because we adopt the RoPE scaling method described in Polaris (An et al., 2025).

## C Experimental Details

In this appendix, we provide the details of our main experiments in Section 4.

### C.1 Training Details

We build on the VeRL codebase (Sheng et al., 2024), with the RLVR loss following the DAPO recipe (Yu et al., 2025). Specifically, the RLVR loss is defined in eq. (1). For our TFPI, it becomes

$$\mathbb{E}_{x \sim \mathcal{D}} \left[ \mathcal{J}_{\text{DAPO}}(\theta, x') \right], \quad x' = \textit{ThinkingFree}(x). \tag{6}$$

For fair comparison, we use identical hyperparameters across methods. For *clip-higher*, we set $\varepsilon_{\text{low}} = 0.2$ and $\varepsilon_{\text{high}} = 0.28$. Training is performed with a batch size and mini-batch size of 256, a learning rate of $10^{-6}$, and no warm-up scheduling. Both KL divergence loss and entropy loss are excluded.

For rollout, we use temperature $= 1$, topp $= 1$, and topk $= -1$. We generate 8 rollouts per problem. Experiments are conducted on `DS-1.5B`, `Qwen3-4B`, and `DS-7B`, with Polaris-53K (An et al., 2025) as the training dataset. **In principle, we could apply dataset filtering at each training stage to accelerate training** An et al. (2025). **However, for fairness, we deliberately use the full training set for all experiments.**

For Direct RLVR, the maximum output length is set to 16K for `DS-1.5B` and `DS-7B`, and 32K for `Qwen3-4B`. For TFPI, we adopt a multi-stage training strategy (An et al., 2025; Luo et al., 2025b):

- `DS-1.5B` and `DS-7B`: 2K $\rightarrow$ 4K $\rightarrow$ 8K.
- `Qwen3-4B`: 4K $\rightarrow$ 8K $\rightarrow$ 16K.

Our experiments are conducted with 32 H20 GPUs. A summary of the number of training steps and training time is provided in Table 6. To provide a comprehensive view of training compute, we also report the number of processed tokens and the average MFU in Table 7. Notably, our method achieves a higher MFU and therefore can generate more tokens than "Direct RL" within the same wall-clock time. This arises because shorter responses during RL training reduce rollout-stage "bubble", yielding a practical efficiency advantage.

### C.2 Baselines

To evaluate the efficacy of TFPI, we compare TFPI with direct RLVR training from an SFT-distilled LRM ("Direct RL" for short) under the same total training compute, i.e., the combined compute of the three TFPI stages equals that of direct RLVR (Table 1). To examine the effect of inserting a TFPI

Table 6: Training Steps and Time of Main Experiments. "kh" denotes one thousand H20 Hours.

| | DS-1.5B | Qwen3-4B | DS-7B | DS-1.5B | Qwen3-4B | DS-7B |
|---|---|---|---|---|---|---|
| TFPI Stage 1 | 2K, 1K steps | 4K, 100 steps | 2K, 1K steps | 0.84 kh | 0.35 kh | 1.73 kh |
| TFPI Stage 2 | 4K, 440 steps | 8K, 56 steps | 4K, 232 steps | 0.62 kh | 0.52 kh | 0.82 kh |
| TFPI Stage 3 | 8K, 440 steps | 16K, 64 steps | 8K, 144 steps | 1.21 kh | 0.91 kh | 0.94 kh |
| TFPI Total | - | - | - | 2.66 kh | 1.79 kh | 3.50 kh |
| Direct RLVR | 16K, 456 steps | 32K, 20 steps | 16K, 280 steps | 2.67 kh | 1.9 kh | 3.52 kh |
| TFPI: RLVR | 16K, 472 steps | 32K, 192 steps | 536 steps | 2.49 kh | 13.5 kh | 6.55 kh |
| TFPI + RLVR Total | - | - | - | 5.16 kh | 15.4 kh | 10 kh |
| Direct RLVR Total | 16K, 896 steps | 32K, 216 steps | 16K, 820 steps | 5.17 kh | 15.7 kh | 10.1 kh |

Table 7: Total processed tokens and the average MFU (in %) of Main Experiments.

| | DS-1.5B | Qwen3-4B | DS-7B | DS-1.5B | Qwen3-4B | DS-7B |
|---|---|---|---|---|---|---|
| TFPI Stage 1 | $2.37 \times 10^9$ | $5.00 \times 10^8$ | $2.75 \times 10^9$ | 69.27 | 66.28 | 73.73 |
| TFPI Stage 2 | $2.26 \times 10^9$ | $5.02 \times 10^8$ | $1.20 \times 10^9$ | 69.54 | 66.67 | 73.79 |
| TFPI Stage 3 | $3.92 \times 10^9$ | $4.94 \times 10^8$ | $1.24 \times 10^9$ | 68.66 | 66.76 | 73.76 |
| TFPI Total | $8.55 \times 10^9$ | $1.50 \times 10^9$ | $5.19 \times 10^9$ | - | - | - |
| Direct RLVR | $7.48 \times 10^9$ | $6.48 \times 10^8$ | $4.77 \times 10^9$ | 68.42 | 61.57 | 72.83 |
| TFPI: RLVR | $6.26 \times 10^9$ | $5.80 \times 10^9$ | $7.98 \times 10^9$ | 66.20 | 60.08 | 72.58 |
| TFPI + RLVR Total | $1.48 \times 10^{10}$ | $7.30 \times 10^9$ | $1.27 \times 10^{10}$ | - | - | - |
| Direct RLVR Total | $1.44 \times 10^{10}$ | $7.36 \times 10^9$ | $1.38 \times 10^{10}$ | 68.11 | 62.24 | 72.89 |

stage before RLVR, we apply TFPI to an SFT-distilled model ("TFPI + RL"), continue with standard RLVR, and compare the results with "Direct RL" using approximately the same training compute (Table 2). We also include several high-performing LRMs of the same model size from previous works for reference, including Polaris (An et al., 2025), DeepScaleR (Luo et al., 2025b), Skywork-OR1 (He et al., 2025), and AReal-RL. For both Table 1 and Table 2, all models are evaluated in thinking mode (see also Appendix C.3). To assess the impact of TFPI on reasoning efficiency, we compare our TFPI-trained model with various RL-based efficient reasoning baselines, including TLMRE (Arora & Zanette, 2025), AdaptThink (Zhang et al., 2025b), AutoThink (Tu et al., 2025), Laser (Liu et al., 2025a), L1Max (Aggarwal & Welleck, 2025), and ThinkLess (Fang et al., 2025) (Table 3). The training and testing settings of these baselines, as reported in their original papers, are summarized in Table 8. **For fair comparison, we standardize the testing parameters to `topp` = 0.95, `topk` = −1, and $T$ = 0.6 with a 32K maximum length**, following the recommendations of DeepSeek-R1 (Guo et al., 2025).

Table 8: Training and evaluation details of efficient reasoning baselines from original papers. We unify the evaluation setting for fair comparison in Table 3. Details are provided in Appendix C.2 and C.3.

| Model | Training Data | Training Length | Test Length | Evaluation Details |
|---|---|---|---|---|
| **DeepSeek-Distill-Qwen-1.5B** | | | | |
| TLMRE-DS-1.5B ($\alpha = 0.1$) | 3.2K examples from NuminaMath | – | 32K | 10 generations for AIME24 |
| AdaptThink-1.5B ($\delta = 0.05$) | DeepScaleR-40K | 16K | 16K | $T = 0.6$, 16 generations for AIME24 |
| AutoThink-DS-1.5B-Stage3 | DeepScaleR-40K | 8K → 16K → 24K | – | $T = 0.6$, 16 generations |
| Laser-DE-L4096-1.5B | DeepScaleR-40K | – | 32K | 16 generations for AIME24 |
| **DeepscaleR-1.5B-Preview** | | | | |
| AutoThink-Stage3 | DeepScaleR-40K | 8K → 16K → 24K | – | $T = 0.6$, 16 generations |
| L1-1.5B-Max | DeepScaleR-40K | 4K | 8K | – |
| Thinkless-1.5B-RL | DeepScaleR-40K | 24K | – | – |

## C.3 EVALUATION DETAILS

To comprehensively evaluate model capabilities, we employ a diverse set of benchmarks covering mathematical reasoning, multi-task reasoning, code generation, and instruction-following:

1. **Mathematical reasoning:** We evaluate on AIME24, AIME25, and BeyondAIME (ByteDance-Seed, 2025). For AIME24 and AIME25, we report `pass@1` accuracy using 32 samples per problem (`avg@32`); for BeyondAIME, we report `avg@8`.

2. **Multi-task reasoning:** We evaluate on GPQA-Diamond (Rein et al., 2024) and report `pass@1` with 8 samples per problem.

3. **Code generation:** We assess coding ability on LiveCodeBench (Jain et al., 2024) (2024-08–2025-01 subset, aligned with DeepSeek-R1 (Guo et al., 2025)), reporting `pass@1` with 8 samples per problem.

4. **Instruction-following:** We evaluate on IFEval (Zhou et al., 2023) and report `pass@1` of the strict prompt accuracy with 4 samples per problem.

All evaluation codes are adapted from the DeepscaleR (Luo et al., 2025b) codebase, where vLLM (Kwon et al., 2023) is leveraged to accelerate inference. For IFEval, we use the same codes provided by the official paper (Zhou et al., 2023).

We provide our decoding parameters as follows:

- **Table 1**: We set the temperature to 0.6 and `topp` = 0.95. For LRMs trained from `DS-1.5B` and `DS-7B`, we use `topk` = −1 with a maximum sequence length of 32K tokens. For LRMs trained from `Qwen3-4B`, we use `topk` = 20 with a maximum sequence length of 48K tokens, applying RoPE scaling as proposed by An et al. (2025).

- **Table 2**: We set the temperature to 0.6 and `topp` = 0.95. For LRMs initialized from `DS-1.5B` and `DS-7B`, we use `topk` = −1 with a maximum sequence length of 32K tokens. For LRMs initialized from `Qwen3-4B`, we use `topk` = 20 with a maximum sequence length of 48K tokens, again applying RoPE scaling as proposed by An et al. (2025). For models marked with "*", we report the results from their original publications (see Appendix C.4).

- **Table 3**: For efficient reasoning baselines listed in Table 8 and the thinking mode of initial models, we set `topp` = 0.95, `topk` = −1, and $T = 0.6$, with a maximum sequence length of 32K tokens (48K for `Qwen3-4B`). For the thinking-free mode of initial models and our TFPI, we set `topp` = 0.8, `topk` = 20, and $T = 0.7$ with a maximum sequence length of 32K tokens, following Yang et al. (2025a).

### C.4 SOURCE OF SOME RESULTS IN TABLE 2

Results for LRMs marked with "*" are taken directly from the Seed-1.5-Thinking report (Seed et al., 2025) and the corresponding Hugging Face page of Qwen3-2507. Note that the LiveCodeBench test set subsets of these results, and the metric of IFEval may differ from those in our experiments; their results are included for reference only.

## D ANALYSIS DETAILS

As *verification* is an important indicator of slow-thinking capabilities (Setlur et al., 2025), we conduct experiments in Section 5.1 to examine how *verification* can generalize to slow-thinking, even when trained with TFPI (thinking-free mode). Following Yang et al. (2025b), for the experiments in Figure 3, we segmented the reasoning trajectories using '\n\n' as delimiters and classified each step according to whether it contained verification-related phrases such as "wait", "let me verify", "let me check", "checking", "verifying", or "double-check".

## E THE USE OF LARGE LANGUAGE MODELS

We only used LLMs (specifically GPT-5) for language refinement during the writing process of this work; all ideas, experimental designs, and analyses were conducted by the authors. The text polished by GPT-5 was subsequently reviewed and verified by us to ensure the quality of the content.

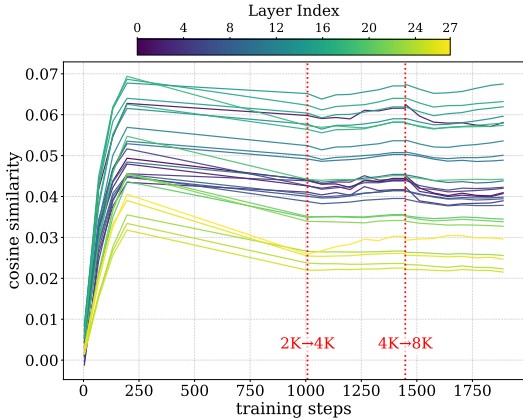

Figure 6: Cosine similarity between parameter updates of TFPI-trained checkpoints and an early "Direct RL" update of `DS-1.5B` across layers during training.

## F  MORE RESULTS

### F.1  ADDITIONAL RESULTS OF PARAMETER-LEVEL ANALYSIS

Figure 4 (right) reports the cosine similarity of parameter-update vectors between TFPI-trained checkpoints and the final Direct RL checkpoint. For completeness, Figure 6 presents the cosine similarity between TFPI-trained checkpoints and an early Direct RL checkpoint (step 456) from the `DS-1.5B` run used in Table 1. We observe stronger alignment with the final Direct RL checkpoint than with the early-stage checkpoint.

### F.2  COMPLETE RESULTS OF TABLE 2 AND TABLE 3

Due to the page limit, we present only representative results in Sections 4.3 and 4.4, with the complete results provided in Tables 9 and 10, respectively.

### F.3  THINKING VS. THINKING-FREE INFERENCE

To compare thinking and thinking-free modes, Table 11 reports accuracy and average output token counts across training stages of TFPI for `Qwen3-4B`. The accuracy gap between the two modes narrows from Stage 1 to Stage 3. In Stage 3, the thinking mode achieves an average accuracy of 63.9%, whereas the thinking-free mode achieves 60.0%. Meanwhile, the thinking-free mode produces substantially fewer tokens on average (9.9k vs. 15.3k). One may choose the thinking-free mode to trade a modest decrease in accuracy for improved efficiency, or switch between modes to balance accuracy and cost.

### F.4  ABOUT DECODING PARAMETERS

The decoding parameters used in our main-text experiments are detailed in Appendices B and C.3. Following the official Hugging Face recommendations for each model, we adopt different settings for the thinking and thinking-free modes. To assess robustness, we additionally report results under a unified configuration: temperature = 0.6, top-p = 0.95, top-k = 20, and a maximum output length of 32k tokens. As shown in Tables 13, 14, 15, and 16, all of our conclusions remain unchanged under this unified setting.

Polaris (An et al., 2025) reports that longer-context inference with RoPE scaling improves performance on harder questions and evaluates at 90K tokens. We find that 90K is prohibitively slow, and 48K suffices based on the clip ratio analysis (Table 12). As shown in Table 15, constraining the maximum output length to 32K slightly degrades the performance of the thinking mode for models RL-tuned from `Qwen3-4B`. We therefore recommend a 48K context length for `Qwen3-4B`-based RL-tuned models.

Table 9: Results (%) of RL after TFPI ("TFPI+RL") vs. "Direct RL" across different benchmarks. "Avg@k" denotes the average accuracy (%) over $k$ random generations (i.e., pass@1). For LRMs marked with "*", results are taken from the corresponding reports (see Appendix C.4); all other results are from our own runs. All models are evaluated in thinking mode. The total training compute for "TFPI+RL" is matched to that of "Direct RL" for fair comparison.

| Models | Mathematics | | | Multi-Task | Code | Instruction | Overall |
|---|---|---|---|---|---|---|---|
| | AIME 24 Avg@32 | AIME 25 Avg@32 | Beyond AIME Avg@8 | GPQA Avg@8 | LiveCode Avg@8 | IFEval Avg@4 | Overall Avg. |
| DeepSeek R1* | 79.8 | 65.0 | 42.4 | 71.5 | 64.3 | 86.1 | 68.2 |
| Seed-1.5-Thinking* | 86.7 | 74.0 | 48.0 | 77.3 | 64.9 | 87.4 | 73.0 |
| Claude4 Opus Thinking* | - | 75.5 | - | 79.6 | 48.9 | 89.7 | - |
| Qwen3-235B-Thinking* | 85.7 | 81.5 | - | 71.1 | 55.7 | 83.4 | - |
| Qwen3-8B | 75.3 | 67.0 | 43.2 | 61.7 | 51.7 | 66.0 | 61.7 |
| DS-R1-0528-Qwen3-8B | 82.8 | 74.6 | 50.6 | 61.4 | 59.0 | 73.9 | 67.0 |
| **1.5B Size** | | | | | | | |
| DeepScaleR-1.5B | 37.8 | 31.6 | 13.1 | 19.1 | 21.9 | 40.5 | 27.3 |
| DS-1.5B Direct RL | 37.2 | 28.5 | 12.9 | 24.6 | 19.5 | 39.5 | 27.0 |
| DS-1.5B TFPI stage 3 | 40.1 | 30.8 | 13.8 | 29.6 | 19.9 | 40.8 | 29.2 |
| DS-1.5B TFPI + RL | 42.3 | 32.9 | 15.1 | 27.8 | 20.9 | 41.0 | 30.0 |
| **4B Size** | | | | | | | |
| Polaris-4B-Preview | 73.2 | 70.7 | 39.9 | 54.9 | 39.7 | 63.9 | 57.0 |
| Qwen3-4B Direct RL | 78.8 | 71.5 | 46.4 | 56.2 | 54.3 | 65.1 | 62.0 |
| Qwen3-4B TFPI stage 3 | 79.9 | 70.6 | 46.7 | 58.5 | 57.0 | 70.2 | 63.8 |
| Qwen3-4B TFPI + RL | 80.8 | 76.0 | 49.5 | 61.1 | 55.7 | 71.3 | 65.7 |
| Qwen3-4B-2507-Thinking | 84.7 | 79.2 | 51.5 | 66.4 | 62.4 | 68.0 | 68.7 |
| - TFPI only | 89.0 | 81.2 | 52.5 | 70.1 | 65.5 | 66.7 | 70.8 |
| **7B Size** | | | | | | | |
| AReal-boba-RL-7B | 61.5 | 46.1 | 30.7 | 35.4 | 34.2 | 57.5 | 44.2 |
| Skywork-OR1-7B-Preview | 61.2 | 46.4 | 31.2 | 35.2 | 43.3 | 55.4 | 45.5 |
| Polaris-7B-Preview | 70.6 | 48.8 | 37.0 | 34.1 | 43.9 | 55.6 | 48.3 |
| DS-7B Direct RL | 62.3 | 47.9 | 30.1 | 36.9 | 42.8 | 57.1 | 46.2 |
| DS-7B TFPI stage 3 | 62.0 | 44.6 | 31.1 | 46.8 | 42.1 | 60.2 | 47.8 |
| DS-7B TFPI + RL | 65.3 | 47.1 | 33.2 | 45.9 | 43.3 | 57.3 | 48.7 |

Table 10: Comparison of the Thinking-Free inference mode of TFPI with efficient reasoning baselines across various reasoning tasks. "Avg@k" denotes the average accuracy (in %) over k generations (i.e., pass@1), and "Toks" indicates the average output length in thousands of tokens (K).

| Models | AIME 24 | | AIME 25 | | Beyond AIME | | GPQA | | Overall | |
|---|---|---|---|---|---|---|---|---|---|---|
| | avg@32 | Toks | avg@32 | Toks | avg@8 | Toks | avg@8 | Toks | Avg. | Toks |
| **DeepSeek-Distill-Qwen-1.5B** | | | | | | | | | | |
| TLMRE-DS-1.5B ($\alpha = 0.1$) | 27.6 | 12.9 | 24.8 | 12.5 | 9.2 | 11.9 | 14.8 | 7.5 | 19.1 | 11.2 |
| AdaptThink-1.5B ($\delta = 0.05$) | 28.1 | 8.0 | 22.6 | 7.9 | 10.0 | 4.8 | 14.8 | 5.1 | 18.9 | 6.5 |
| AutoThink-DS-1.5B-Stage3 | 30.3 | 10.2 | 25.2 | 9.1 | 9.4 | 8.7 | 17.4 | 7.0 | 20.6 | 8.7 |
| Laser-DE-L4096-1.5B | 30.3 | 8.3 | 24.9 | 7.4 | 9.7 | 7.4 | 21.1 | 4.7 | 21.5 | 6.9 |
| Initial Model (Thinking) | 29.6 | 16.7 | 23.0 | 16.5 | 8.7 | 14.4 | 16.3 | 9.8 | 19.4 | 14.3 |
| Initial Model (Thinking-Free) | 12.4 | 5.7 | 10.9 | 4.4 | 4.4 | 3.4 | 4.2 | 0.9 | 8.0 | 3.6 |
| - TFPI stage 1 | 21.9 | 1.6 | 15.3 | 1.4 | 8.7 | 1.3 | 32.9 | 0.8 | 19.7 | 1.3 |
| - TFPI stage 2 | 31.5 | 3.4 | 24.2 | 3.1 | 10.1 | 2.9 | 35.3 | 1.6 | 25.3 | 2.7 |
| - TFPI stage 3 | 37.5 | 5.3 | 28.4 | 5.0 | 12.4 | 4.9 | 35.6 | 2.6 | 28.5 | 4.4 |
| **DeepscaleR-1.5B-Preview** | | | | | | | | | | |
| AutoThink-Stage3 | 38.9 | 8.7 | 28.9 | 7.7 | 11.6 | 7.8 | 27.3 | 5.4 | 26.7 | 7.4 |
| L1-1.5B-Max | 27.2 | 3.2 | 26.3 | 2.9 | 9.1 | 3.1 | 32.4 | 2.3 | 23.8 | 2.9 |
| Thinkless-1.5B-RL | 28.4 | 11.3 | 24.1 | 11.1 | 8.1 | 11.7 | 20.3 | 12.7 | 20.2 | 11.7 |
| **Qwen3-4B** | | | | | | | | | | |
| Initial Model (Thinking) | 73.6 | 18.0 | 68.3 | 22.3 | 43.4 | 23.5 | 56.8 | 10.7 | 60.5 | 18.6 |
| Initial Model (Thinking-Free) | 26.9 | 7.2 | 20.2 | 5.5 | 11.0 | 4.2 | 45.0 | 2.3 | 25.8 | 4.8 |
| - TFPI stage 1 | 43.9 | 3.7 | 31.1 | 3.4 | 23.2 | 3.4 | 47.8 | 1.5 | 36.5 | 3.0 |
| - TFPI stage 2 | 63.7 | 7.2 | 52.8 | 8.0 | 33.0 | 7.0 | 50.9 | 2.3 | 50.1 | 6.1 |
| - TFPI stage 3 | 75.1 | 10.5 | 66.9 | 12.7 | 42.1 | 12.7 | 55.9 | 3.7 | 60.0 | 9.9 |
| Qwen3-4B-Instruct-2507 | 60.4 | 7.9 | 46.1 | 7.4 | 33.0 | 7.3 | 64.4 | 4.8 | 51.0 | 6.9 |

Table 11: Comparison of the Thinking-Free inference mode with the Thinking mode across various reasoning tasks after TFPI. "`Avg@k`" denotes the average accuracy (in %) over k generations (i.e., `pass@1`), and "`Toks`" indicates the average output length in thousands of tokens (K).

| Models | AIME 24 | | AIME 25 | | Beyond AIME | | GPQA | | Overall | |
|---|---|---|---|---|---|---|---|---|---|---|
| | avg@32 | Toks | avg@32 | Toks | avg@8 | Toks | avg@8 | Toks | Avg. | Toks |
| **Qwen3-4B** | | | | | | | | | | |
| Initial Model (Thinking) | 73.6 | 18.0 | 68.3 | 22.3 | 43.4 | 23.5 | 56.8 | 10.7 | 60.5 | 18.6 |
| Initial Model (Thinking-Free) | 26.9 | 7.2 | 20.2 | 5.5 | 11.0 | 4.2 | 45.0 | 2.3 | 25.8 | 4.8 |
| - TFPI stage 1 (Thinking) | 75.2 | 13.8 | 67.8 | 17.1 | 42.4 | 16.8 | 57.9 | 7.5 | 60.8 | 13.8 |
| - TFPI stage 1 (Thinking-Free) | 43.9 | 3.7 | 31.1 | 3.4 | 23.2 | 3.4 | 47.8 | 1.5 | 36.5 | 3.0 |
| - TFPI stage 2 (Thinking) | 76.0 | 13.4 | 68.2 | 17.0 | 44.7 | 16.7 | 57.8 | 7.7 | 61.7 | 13.7 |
| - TFPI stage 2 | 63.7 | 7.2 | 52.8 | 8.0 | 33.0 | 7.0 | 50.9 | 2.3 | 50.1 | 6.1 |
| - TFPI stage 3 (Thinking) | 79.9 | 15.1 | 70.6 | 18.7 | 46.7 | 18.8 | 58.5 | 8.6 | 63.9 | 15.3 |
| - TFPI stage 3 | 75.1 | 10.5 | 66.9 | 12.7 | 42.1 | 12.7 | 55.9 | 3.7 | 60.0 | 9.9 |
| Qwen3-4B-Instruct-2507 | 60.4 | 7.9 | 46.1 | 7.4 | 33.0 | 7.3 | 64.4 | 4.8 | 51.0 | 6.9 |

| Response Length | 16K | 32K | 48K | 64K | 80K | 90K |
|---|---|---|---|---|---|---|
| Avg@32 | 42.6 | 64.1 | 68.3 | 68.5 | 68.5 | 68.5 |
| Clip Ratio | 55.7 | 25.7 | 5.2 | 1.2 | 0.6 | 0.0 |

Table 12: AIME25 results for `Qwen3-4B`: `Avg@32` and clip ratio across maximum output lengths.

## F.5 ABOUT TRAINING TEMPERATURE

In our main experiments, we set the training temperature to 1.0 (Appendix C.1). Motivated by Polaris(An et al., 2025), which reports an "optimal" temperature of 1.4 for `Qwen3-4B`, we conduct an ablation on training temperature using `Qwen3-4B`. As shown in Figure 7, performance at 1.4 is comparable to 1.0 for both TFPI Stage 1 and Direct RL (measured by `AVG@32` on AIME25), suggesting that our results are robust to this choice of training temperature.

## F.6 ABOUT THE STAGE BOUNDARY AND STAGE LENGTH

Following prior multi-stage training practice (An et al., 2025; Luo et al., 2025b), we increase the training context length when the clip ratio shows a renewed inflection and validation performance begins to plateau. Figure 8 illustrates this behavior for TFPI Stage 1 on `Qwen3-4B`; step 100 is a natural transition point.

Training context length trades off training throughput and effective learning signal: longer contexts slow training, whereas shorter contexts constrain learning on harder questions. We therefore begin with a shorter length and expand as training stabilizes. To set stage lengths, we prioritize effective signal: with 8-sample rollouts, 0/8 or 8/8 correctness contributes no learning signal (Yu et al., 2025), so for a candidate length L, we compute the share of questions with 1/8–7/8 correctness as a proxy for effective training mass. We choose L to preserve throughput while maintaining substantial effective mass at each stage. For `Qwen3-4B`, L=4K yields 28K/53K questions in the 1/8–7/8 band, whereas L=8K yields 29K/53K; thus we start at 4K for efficiency and move to 8K once validation gains plateau. Note that the effective sets across stages are not disjoint; for instance, the same ques-

Table 13: Re-evaluate the meta-experiment in Figure 2 (Left) under a unified evaluation setting (temperature = 0.6, top-p = 0.95, top-k = 20). "# Tokens" denotes the average number of tokens on AIME25 and "$\Delta$" is the percentage of reduction in token consumption.

| Model | # Tokens (Thinking) | # Tokens (Thinking-Free) | $\Delta$ |
|---|---|---|---|
| `DS-1.5B` (original setting) | 16.5K | 4.4K | -73% |
| Unified Setting | 16.4K | 3.6K | -78% |
| `Qwen3-4B` (original setting) | 17.4K | 5.1K | -71% |
| Unified Setting | 17.4K | 4.7K | -73% |

Table 14: Results of TFPI vs. direct RL across different benchmarks. "Avg@k" denotes the average accuracy (%) over $k$ random generations (i.e., pass@1). All models are evaluated in thinking mode. The total training compute for the 3 stages of TFPI equals that of "Direct RL" for fair comparison. This table corresponds to Table 1 under a unified decoding setting.

| Models | Mathematics | | | Multi-Task | Code | Instruction | Overall |
|---|---|---|---|---|---|---|---|
| | AIME 24 Avg@32 | AIME 25 Avg@32 | Beyond AIME Avg@8 | GPQA Avg@8 | LiveCode Avg@8 | IFEval Avg@4 | Overall Avg. |
| **DeepSeek-Distill-Qwen-1.5B** | | | | | | | |
| Initial Model | 29.1 | 25.5 | 9.0 | 16.3 | 17.6 | 40.1 | 22.9 |
| - Direct RL | 35.5 | 27.9 | 12.5 | 20.2 | 18.7 | 38.1 | 25.5 |
| - TFPI stage 1 | 31.8 | 26.7 | 12.5 | 30 | 18.9 | 39.5 | 26.6 |
| - TFPI stage 2 | 35.1 | 27.6 | 13.4 | 29.7 | 18.9 | 44.0 | 28.1 |
| - TFPI stage 3 | 36.1 | 30.1 | 14.9 | 28.5 | 21.0 | 41.8 | 28.7 |
| **Qwen3-4B** | | | | | | | |
| Initial Model | 71.6 | 64.1 | 41.5 | 56.6 | 54.9 | 64.9 | 58.9 |
| Direct RL | 73.4 | 62.2 | 41.1 | 56.2 | 52.1 | 66.0 | 58.5 |
| TFPI stage 1 | 74.7 | 67.4 | 42.1 | 57.9 | 55.3 | 66.0 | 60.6 |
| TFPI stage 2 | 75.6 | 67.5 | 44.6 | 57.8 | 54.6 | 64.8 | 60.8 |
| TFPI stage 3 | 78.7 | 69.5 | 46.1 | 58.5 | 56.8 | 70.2 | 63.3 |
| **DeepSeek-Distill-Qwen-7B** | | | | | | | |
| Initial Model | 53.3 | 40.5 | 25.0 | 36.8 | 37.6 | 55.6 | 41.5 |
| - Direct RL | 56.8 | 41.6 | 26.7 | 37.4 | 39.6 | 58.2 | 43.4 |
| - TFPI stage 1 | 58.8 | 39.6 | 28.8 | 49.4 | 40.1 | 56.6 | 45.5 |
| - TFPI stage 2 | 61.7 | 42.9 | 31.5 | 47.5 | 42.0 | 57.5 | 47.2 |
| - TFPI stage 3 | 62.6 | 43.2 | 30.7 | 47.1 | 42.4 | 59.3 | 47.5 |

Table 15: Results (%) of RL after TFPI ("TFPI+RL") vs. "Direct RL" across different benchmarks. "Avg@k" denotes the average accuracy (%) over $k$ random generations (i.e., pass@1). For LRMs marked with "*", results are taken from the corresponding reports (see Appendix C.4); All models are evaluated in thinking mode. The total training compute for "TFPI+RL" is matched to that of "Direct RL" for fair comparison. This table corresponds to Table 2 under a unified decoding setting.

| Models | Mathematics | | | Multi-Task | Code | Instruction | Overall |
|---|---|---|---|---|---|---|---|
| | AIME 24 Avg@32 | AIME 25 Avg@32 | Beyond AIME Avg@8 | GPQA Avg@8 | LiveCode Avg@8 | IFEval Avg@4 | Overall Avg. |
| DeepSeek R1* | 79.8 | 65.0 | 42.4 | 71.5 | 64.3 | 86.1 | 68.2 |
| Seed-1.5-Thinking* | 86.7 | 74.0 | 48.0 | 77.3 | 64.9 | 87.4 | 73.0 |
| Claude4 Opus Thinking* | - | 75.5 | - | 79.6 | 48.9 | 89.7 | - |
| Qwen3-235B-Thinking* | 85.7 | 81.5 | - | 71.1 | 55.7 | 83.4 | - |
| Qwen3-4B Direct RL | 75.8 | 65.8 | 42.1 | 56.0 | 53.9 | 65.1 | 59.8 |
| Qwen3-4B TFPI stage 3 | 78.7 | 69.5 | 46.1 | 58.5 | 56.8 | 70.2 | 63.3 |
| Qwen3-4B TFPI + RL | 77.9 | 70.8 | 47.5 | 60.8 | 54.5 | 71.3 | 63.8 |

tion may score 3/8 at 4K and 6/8 at 8K. In all experiments, we train on the full Polaris-53K dataset to ensure fairness. However, for better efficiency, one can curate stage-specific subsets based on the effective-signal criterion by filtering out questions that are 0/8 or 8/8 correct under the current context length.

## F.7 MULTI-STAGE DIRECT RL (4K → 8K → 16K)

Figure 2 (right) shows that applying Direct RL with a 4K response length causes a substantial performance drop on AIME25 for Qwen3-4B. In contrast, TFPI trains stably even at 4K, suggesting that—even in a multi-stage setting—the initial stage length should not be too short. We posit that enabling multi-stage training with short response lengths is a key benefit of TFPI. To rule out the possibility that this effect is an artifact of staging, we run Direct RL with the identical multi-stage schedule on Qwen3-4B. The complete results are reported in Table 17, with training details provided in Table 18.

Table 16: Comparison of the Thinking-Free inference mode of TFPI with efficient reasoning baselines across various reasoning tasks. "Avg@k" denotes the average accuracy (in %) over k generations (i.e., pass@1), and "Toks" indicates the average output length in thousands of tokens (K). Models with "*" are trained from DeepScaleR-1.5B, while the remaining are from DS-1.5B. This table corresponds to Table 3 under a unified decoding setting.

| Models | AIME 24 | | AIME 25 | | Beyond AIME | | GPQA | | Overall | |
|---|---|---|---|---|---|---|---|---|---|---|
| | avg@32 | Toks | avg@32 | Toks | avg@8 | Toks | avg@8 | Toks | Avg. | Toks |
| 1) TLMRE-DS-1.5B ($\alpha = 0.1$) | 27.6 | 12.9 | 24.8 | 12.5 | 9.2 | 11.9 | 14.8 | 7.5 | 19.1 | 11.2 |
| 2) AdaptThink-1.5B ($\delta = 0.05$) | 28.1 | 8.0 | 22.6 | 7.9 | 10.0 | 4.8 | 14.8 | 5.1 | 18.9 | 6.5 |
| 3) AutoThink-DS-1.5B-Stage3 | 30.3 | 10.2 | 25.2 | 9.1 | 9.4 | 8.7 | 17.4 | 7.0 | 20.6 | 8.7 |
| 4) Laser-DE-L4096-1.5B | 30.3 | 8.3 | 24.9 | 7.4 | 9.7 | 7.4 | 21.1 | 4.7 | 21.5 | 6.9 |
| 5) AutoThink-Stage3* | 38.9 | 8.7 | 28.9 | 7.7 | 11.6 | 7.8 | 27.3 | 5.4 | 26.7 | 7.4 |
| 6) L1-1.5B-Max* | 27.2 | 3.2 | 26.3 | 2.9 | 9.1 | 3.1 | 32.4 | 2.3 | 23.8 | 2.9 |
| 7) Thinkless-1.5B-RL* | 28.4 | 11.3 | 24.1 | 11.1 | 8.1 | 11.7 | 20.3 | 12.7 | 20.2 | 11.7 |
| DS-1.5B (Thinking) | 29.1 | 16.5 | 25.5 | 16.4 | 9.0 | 17.3 | 16.3 | 9.8 | 20.0 | 15.0 |
| DS-1.5B (Thinking-Free) | 13.3 | 5.1 | 11.0 | 3.6 | 4.0 | 2.6 | 6.2 | 0.9 | 8.6 | 3.0 |
| - TFPI stage 1 | 23.9 | 1.5 | 16.2 | 1.4 | 9.1 | 1.2 | 36.4 | 0.8 | 21.4 | 1.2 |
| - TFPI stage 2 | 28.6 | 3.4 | 25.0 | 3.0 | 10.6 | 2.9 | 35.7 | 1.6 | 25.0 | 2.7 |
| - TFPI stage 3 | 37.7 | 5.2 | 28.4 | 5.1 | 14.5 | 4.9 | 34.1 | 2.6 | 28.7 | 4.4 |

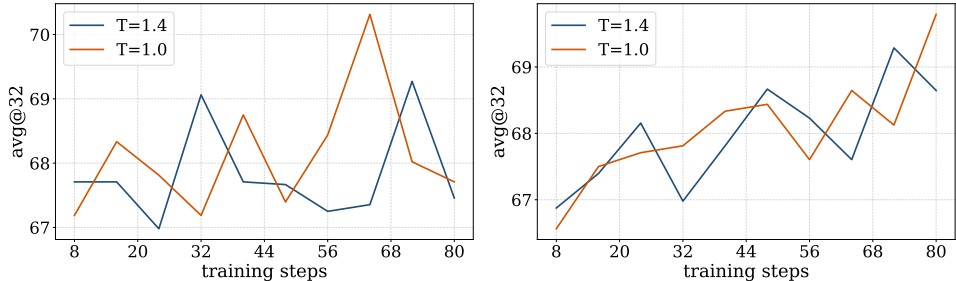

Figure 7: AVG@32 (%) on AIME25 for Qwen3-4B across training steps under different training temperatures (1.4 vs. 1.0). Left: Stage 1 of TFPI; Right: Direct RL.

Table 17: TFPI v.s. Multi-Stage Direct RL. All models are evaluated in thinking mode.

| Models | Mathematics | | | Multi-Task | Code | Instruction | Overall |
|---|---|---|---|---|---|---|---|
| | AIME 24 Avg@32 | AIME 25 Avg@32 | Beyond AIME Avg@8 | GPQA Avg@8 | LiveCode Avg@8 | IFEval Avg@4 | Overall Avg. |
| Initial Model | 73.6 | 68.3 | 43.4 | 56.8 | 54.9 | 64.9 | 60.3 |
| TFPI stage 1 | 75.2 | 67.8 | 42.4 | 57.9 | 55.3 | 66.0 | 60.8 |
| Multi-Stage Direct RL Stage 1 | 49.2 | 36.0 | 22.6 | 51.4 | 44.0 | 64.5 | 44.6 |
| TFPI stage 2 | 76.0 | 68.2 | 44.7 | 57.8 | 54.8 | 64.8 | 61.0 |
| Multi-Stage Direct RL Stage 2 | 58.4 | 42.3 | 28.2 | 52.9 | 48.4 | 63.2 | 48.9 |
| TFPI stage 3 | 79.9 | 70.6 | 46.7 | 58.5 | 57.0 | 70.2 | 63.8 |
| Multi-Stage Direct RL Stage 3 | 64.9 | 48.6 | 32.5 | 55.2 | 51.0 | 65.2 | 52.9 |

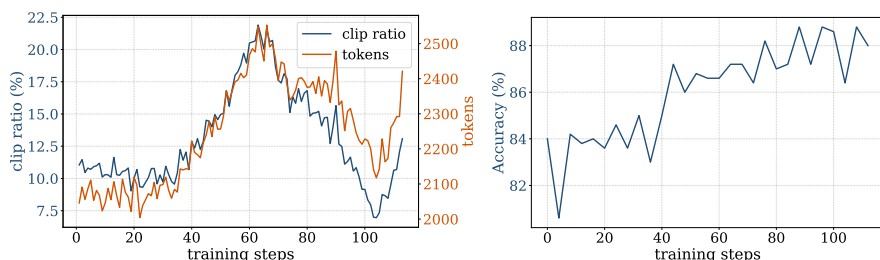

Figure 8: Qwen3-4B TFPI Stage 1. Left: clip ratio and average output tokens over training steps; Right: MATH500 validation accuracy. Step 100 marks a turning point.

Table 18: Training Steps, Time, and MFU of TFPI v.s. Direct RL with `Qwen3-4B`.

| | Training Steps | Training Time (kh) | MFU (%) |
|---|---|---|---|
| TFPI Stage 1 | 4K, 100 steps | 0.35 | 66.28 |
| Multi-Stage Direct RL Stage 1 | 4K, 100 steps | 0.45 | 66.82 |
| TFPI Stage 2 | 8K, 56 steps | 0.52 | 66.67 |
| Multi-Stage Direct RL Stage 2 | 8K, 56 steps | 0.51 | 66.31 |
| TFPI Stage 3 | 16K, 64 steps | 0.91 | 66.76 |
| Multi-Stage Direct RL Stage 3 | 16K, 64 steps | 0.97 | 65.99 |
| TFPI Total | - | 1.79 | - |
| Multi-Stage Direct RL Total | - | 1.93 | - |

## G  REPRODUCIBILITY STATEMENT

To ensure reproducibility, we provide a detailed description of our method in Section 3.3 and present all experimental setups in Section 4.1 and Appendix C, including training configurations and evaluation protocols. All evaluation datasets, the training set, the training framework, testing codes, as well as the efficient reasoning baselines used in our experiments, are publicly available and properly referenced (see Appendix C). We will also release our codes and training checkpoints upon acceptance for reproducibility.

