# OpenReview forum: "Thinking-Free Policy Initialization Makes Distilled Reasoning Models More Effective and Efficient Reasoners"
_ICLR.cc/2026/Conference — ICLR 2026 Poster_

### Official Review · Reviewer_Z381 · 2025-10-21

**Soundness:** 3
**Presentation:** 3
**Contribution:** 3
**Rating:** 6
**Confidence:** 3

**Summary:**

This paper introduces Thinking-Free Policy Initialization (TFPI), a lightweight pre-training stage for reasoning LLMs before RL with verifiable rewards. TFPI removes explicit <think> content, training models to output only final solutions under short contexts. Experiments on DS-1.5B, DS-7B, and Qwen3-4B show that TFPI improves reasoning accuracy while reducing compute. When followed by normal RL, TFPI further enhances performance and speeds up convergence.

**Strengths:**

- Efficiency: TFPI significantly reduces rollout tokens and compute cost while maintaining or improving reasoning accuracy.
- Generality: The method integrates seamlessly with existing RLVR frameworks (e.g., GRPO, DAPO) and scales consistently across 1.5B–7B models.
- Insightfulness: Analysis reveals that TFPI fosters implicit verification-style reasoning and provides better initialization for subsequent long-context RL.

**Weaknesses:**

- The paper adopts a multi-stage TFPI schedule (e.g., 2K→4K→8K→16K) but does not justify how these stage boundaries are chosen. It is unclear whether they are based on empirical tuning, stability constraints, or model-size scaling rules.
- The paper only evaluates TFPI on models up to 7 B parameters. It remains unclear whether the approach scales effectively to larger reasoning models (e.g., 32 B or 14B)

**Questions:**

- what happens if we remove the think segment entirely at inference after training—does accuracy remain similar?
- How are the stage lengths (2K→4K→8K) determined in practice? Are they tuned for stability or fixed heuristically?

---

> ### Author Response · Authors · 2025-11-20
> **Response to Reviewer Z381 [1/2]**
>
> Dear Reviewer Z381,
>
> Thank you for the time and effort you invested in reviewing our work. We are grateful for your positive feedback. For your questions/concerns, we will address them one by one as follows:
>
> ## Weakness 1
> > The paper adopts a multi-stage TFPI schedule (e.g., 2K→4K→8K→16K) but does not justify how these stage boundaries are chosen. It is unclear whether they are based on empirical tuning, stability constraints, or model-size scaling rules.
>
> Thank you for the insightful question. Our stage boundaries are determined by online diagnostics rather than fixed scaling rules. **Concretely, we monitor**:
>
> - Training rollouts: average generated tokens and the clip ratio (fraction of sequences truncated at the current limit).
> - Validation: accuracy trends on a held-out set.
>
> We advance to the next context length when the clip ratio begins to rise again (or remains persistently elevated) and validation accuracy shows clear slowing/plateauing, indicating that the current limit is constraining learning.
> This balances training stability with throughput: we keep the context as short as possible until truncation starts to hinder progress.
>
> **We added Appendix F.4 detailing this discussion and Figure 8 illustrating the dynamics in our revised paper** (for Qwen3-4B, Stage 1 shows a turning point around step 100). Please see our updated PDF for the figure.
>
> ## Weakness 2
>
> > The paper only evaluates TFPI on models up to 7 B parameters. It remains unclear whether the approach scales effectively to larger reasoning models (e.g., 32 B or 14B)
>
> Thank you for raising this point. We are currently running experiments on Qwen3-14B; as this will take some time, we will update the PDF with the results and leave comments once they are available.
>
> ## Question 1
> > what happens if we remove the think segment entirely at inference after training—does accuracy remain similar?
>
> Thank you for your question! In fact, Table 3 and Table 10 are the results of "removing the think segment entirely" (i.e., thinking-free mode).
>
> **To help compare thinking mode results with thinking-free mode more easily, we have added Table 11 in Appendix F.3 (see our updated PDF)** to report accuracy and average output token counts across training stages of TFPI for Qwen3-4B.
> For your convenience, we also put the table in markdown here:
>
> | Models  | AIME 24 avg@32 | AIME 24 Toks | AIME 25 avg@32 | AIME 25 Toks | Beyond AIME avg@8 | Beyond AIME Toks | GPQA avg@8 | GPQA Toks | Overall Avg. | Overall Toks |
> |------|----------------|--------------|----------------|--------------|-------------------|------------------|------------|-----------|--------------|--------------|
> | Initial Model (Thinking)         | 73.6           | 18.0         | 68.3           | 22.3         | 43.4              | 23.5             | 56.8       | 10.7      | 60.5         | 18.6         |
> | Initial Model (Thinking-Free)    | 26.9           | 7.2          | 20.2           | 5.5          | 11.0              | 4.2              | 45.0       | 2.3       | 25.8         | 4.8          |
> | – TFPI stage 1 (Thinking)      | 75.2           | 13.8         | 67.8           | 17.1         | 42.4              | 16.8             | 57.9       | 7.5       | 60.8         | 13.8         |
> | – TFPI stage 1 (Thinking-Free) | 43.9           | 3.7          | 31.1           | 3.4          | 23.2              | 3.4              | 47.8       | 1.5       | 36.5         | 3.0          |
> | – TFPI stage 2 (Thinking)      | 76.0           | 13.4         | 68.2           | 17.0         | 44.7              | 16.7             | 57.8       | 7.7       | 61.7         | 13.7         |
> | – TFPI stage 2 (Thinking-Free) | 63.7           | 7.2          | 52.8           | 8.0          | 33.0              | 7.0              | 50.9       | 2.3       | 50.1         | 6.1          |
> | – TFPI stage 3 (Thinking)      | 79.9           | 15.1         | 70.6           | 18.7         | 46.7              | 18.8             | 58.5       | 8.6       | 63.9         | 15.3         |
> | – TFPI stage 3 (Thinking-Free) | 75.1           | 10.5         | 66.9           | 12.7         | 42.1              | 12.7             | 55.9       | 3.7       | 60.0         | 9.9          |
> | Qwen3-4B-Instruct-2507           | 60.4           | 7.9          | 46.1           | 7.4          | 33.0              | 7.3              | 64.4       | 4.8       | 51.0         | 6.9          |
>
> From the table, we have:
> The accuracy gap between the two modes narrows from Stage~1 to Stage~3.
> In Stage~3, the thinking mode achieves an average accuracy of 63.9\%, whereas the thinking-free mode achieves 60.0\%. Meanwhile, the thinking-free mode produces substantially fewer tokens on average (9.9k vs.\ 15.3k).

---

> > ### Author Response · Authors · 2025-11-20
> > **Response to Reviewer Z381 [2/2]**
> >
> > ## Question 2
> >
> > > How are the stage lengths (2K→4K→8K) determined in practice? Are they tuned for stability or fixed heuristically?
> >
> > Thanks for the insightful question! The stage lengths are not tuned; they are fixed heuristically.
> > But in practice, we can choose the stage lengths based on the trade-off between throughput and effective learning signal.
> > With 8-sample rollouts, 0/8 and 8/8 correctness provide no learning signal, so for each candidate length L, we can measure the share of questions with 1/8–7/8 correctness as a proxy for effective training mass and choose L to preserve throughput while maintaining substantial effective mass.
> > Take Qwen3-4B for example, 4K yields 28K/53K effective questions versus 29K/53K at 8K; accordingly, we start at 4K for efficiency and move to 8K when validation gains plateau.
> > Note that effective question sets across stages are not disjoint (e.g., a question may score 3/8 at 4K and 6/8 at 8K).
> >
> > For fairness, all our experiments run on the full Polaris-53K.
> > However, one can curate suitable subsets for different stages by filtering out 0/8 and 8/8 questions (under a certain length) beforehand for better training efficiency.
> > **We added this discussion in Appendix F.6 of the revised manuscript.**
> >
> > Thank you again for your thoughtful comments and positive feedback. We hope our responses have addressed your concerns and questions. At this stage, we welcome any follow-up questions or comments and are available to discuss further.
> >
> > Sincerely,
> >
> > TFPI Authors

---

> > ### Comment · Reviewer_Z381 · 2025-11-26
> > **Thanks for your reply**
> >
> > Thank you for the clarification. I have two short follow-up questions regarding Question 1:
> > 1. Have you compared thinking-free training in the no-stage setting (16k) vs. the multi-stage TFPI setting? I am wondering the benefis of the multi-stage setting in thinking-free training.
> > 2. I am also interested in whether thinking-free training shifts part of the reasoning process into the answer segment after training. To examine this in the table of Question1, could you report the frequency of “wait” tokens (or similar reasoning markers) in the generated text, and provide the MATH500 results (accuracy, average output tokens, and “wait” count)? Since AIME and GPQA are very difficult benchmarks, it would be helpful to see the trend on a relatively easier dataset.
> >
> > Thank you again. I would like to raise my score after answering these two questions.

---

> ### Author Response · Authors · 2025-11-23
> **Follow-up Results of 14B Model**
>
> Dear Reviewer Z381,
>
> Thank you for your valuable suggestion to incorporate the results of larger reasoning models (`The paper only evaluates TFPI on models up to 7 B parameters. It remains unclear whether the approach scales effectively to larger reasoning models (e.g., 32 B or 14B)`). **We have finished the experiments of Qwen3-14B and updated the results in Section 5.3 and Table 4**. For your convenience, we give the results as follows:
>
> | Models | AIME24 | AIME25 | Beyond AIME | GPQA | LiveCodeBench | IFEval | Overall|
> | ------ | - | - | - | - | - | - | - |
> | Qwen3-14B | 81.7 | 73.2 | 50.2 | 65.0 | 63.5 | 67.3 | 66.8 |
> | Direct RL | 81.7 | 74.5 | **50.8** | 64.8 | 61.8 | **68.2** | 66.9 |
> | TFPI | **83.2** | **76.0** | 50.5 | **65.6** | **63.8** | 67.8 | **67.8** |
>
> From the table, **TFPI still outperforms Direct RL under the same training compute**. Notably, Direct RL yields results only comparable to those of the initial model (66.9% vs. 66.8% on average). We hypothesize that Polaris-53K is, in
> some sense, too easy for Qwen3-14B (with over 78% rollout accuracy), leaving only approximately 30% effective data after dynamic sampling (excluding 0/8 and 8/8 prompts). Nonetheless, TFPI remains effective in this setting.
>
> Additionally, we also report the average token consumption on test sets as follows (all numbers are in K):
>
> | Models | AIME24 | AIME25 | Beyond AIME | GPQA | LiveCodeBench | Overall|
> | ------ | - | - | - | - | - | - |
> | Qwen3-14B | 16.8 | 20.8 | 22.8 | 9.4 | 16.0 | 17.1 |
> | Direct RL | 17.9 | 22.1 | 24.5 | 10.4 | 17.4 | 18.5 |
> | TFPI | **13.8** | **17.3** | **18.3** | **7.3** | **13.8** | **14.1** |
>
> From the table, **TFPI reduces the average number of output tokens on the test set by 23.8% even in thinking mode** (14.1K vs. 18.5K). This is another merit of TFPI.
>
> These results highlight the effectiveness of TFPI for larger model sizes, and we believe that with more challenging, higher-quality data, TFPI can further realize its scalability potential.
>
> We hope these additional results could help resolve your concern. At this stage, we welcome any follow-up questions or comments and are available to discuss further.
>
> Sincerely,
>
> TFPI Authors

---

> ### Author Response · Authors · 2025-11-27
> **Response to Follow-Up Question [1/2]**
>
> Dear Reviewer Z381,
>
> Thank you for your active engagement. Before addressing the follow‑ups to Question 1, we would like to **clarify the logic of our paper and explain why both inference modes (thinking & thinking-free) appear**:
>
> - **Training Mode. TFPI is trained in thinking‑free mode.** We find that thinking‑free training improves performance when the model is evaluated in thinking mode (Section 3.2).
> - **Performance in thinking mode**. Under thinking‑mode evaluation, TFPI outperforms Direct RL (Section 4.2; Table 1), and TFPI+RL yields further gains (Section 4.3; Table 2).
> - **Efficiency in thinking‑free mode**. TFPI‑trained models are highly token‑efficient when evaluated in thinking‑free mode (Section 4.4; Table 3). Although we remove the entire “thinking” segment at inference, TFPI optimizes the answer segment and tends to lengthen it slightly during training; thus, some reflective behavior may appear within the answer segment. Even so, Table 3 shows our responses are substantially shorter than other efficient‑reasoning baselines.
> - **Comparison to a fast‑thinking baseline**. In our table responding to Question 1, **Qwen3‑4B‑Instruct‑2507 (a fast‑thinking LLM) has comparable average token usage to our Stage‑2 model in thinking‑free mode**.
> - **Stability of the think–answer pattern. As shown in Section 5.2 and Figure 5 (left)**, while thinking‑free inference may contain minor “reflection” within the answer segment, thinking‑mode inference preserves the explicit think–answer structure and does not increase the number of tokens in the answer segment.
>
> We hope this clarifies the rationale behind using both thinking and thinking‑free inference. **In summary, thinking‑free training improves accuracy in thinking mode and improves efficiency in thinking‑free mode.**
>
> > Have you compared thinking-free training in the no-stage setting (16k) vs. the multi-stage TFPI setting? I am wondering the benefis of the multi-stage setting in thinking-free training.
>
> Thank you for your insightful question! We have tried two different stage schedules as suggested by Reviewer dBLU (Question 1):
>
> |Schedule|AIME24|AIME25|Beyond AIME|GPQA|LiveCodeBench|IFEval|Overall|
> |-|-|-|-|-|-|-|-|
> | Initial Model | 73.6 | 68.3 | 43.4 | 56.8 | 54.9 | 64.9 | 60.3 |
> | Direct RL | 75.7 | 67.0 | 43.6 | 56.3 | 52.5 | 66.0 | 60.2 |
> |$4\rm{K}\rightarrow8\rm{K}\rightarrow16\rm{K}$ | 79.9 | 70.6 | 46.7 | 58.5 | 57.0 | 70.2 | 63.8 |
> | $8\rm{K}\rightarrow16\rm{K}$ | 78.0 | 70.2 | 46.7 | 59.5 | 55.8 | 64.5 | 62.4 |
> | $16\rm{K}$ only | 77.1 | 71.5 | 45.8 | 56.4 | 54.4 | 66.0 | 61.9 |
>
> From the table, we have
> + All schedules outperform the Direct RL baseline and achieve comparable performance on math benchmarks, indicating that TFPI is robust to the choice of schedule.
> + $4\mathrm{K}\rightarrow 8\mathrm{K} \rightarrow 16\mathrm{K}$ yields the best overall results (63.8\% vs. 62.4\% and 61.9\%).
>
> We hypothesize that this stems from an implicit data curriculum induced by dynamic sampling: beginning at 4K improves training efficiency and emphasizes easier problems (1/8-7/8 correctness under 4K context length) early on.
>
> We **have incorporated this table and analysis in Section 5.4 in our revised manuscript**.

---

> ### Author Response · Authors · 2025-11-27
> **Response to Follow-Up Question [2/2]**
>
> >I am also interested in whether thinking-free training shifts part of the reasoning process into the answer segment after training. To examine this in the table of Question1, could you report the frequency of “wait” tokens (or similar reasoning markers) in the generated text, and provide the MATH500 results (accuracy, average output tokens, and “wait” count)? Since AIME and GPQA are very difficult benchmarks, it would be helpful to see the trend on a relatively easier dataset.
>
> Thank you for your helpful suggestion! As shown in Section 5.2 and Figure 5 (left), while thinking‑free inference may contain minor “reflection” within the answer segment, thinking‑mode inference preserves the explicit think–answer structure and does not increase the number of tokens in the answer segment.
>
> **As suggested, we provide the results on MATH500 as follows**: (For "reflection words", we use several markers similar to "wait", including "wait", "Hold on", "double-check", etc.)
>
> | Model | MATH500 avg@4 | Tokens (K) | reflection word counts (thinking segment) | reflection word counts (answer segment) |
> | - | - | - | - | - |
> | Initial Model (Thinking) | 94.2 | 6.1 | 33.7 | 0.44 |
> | Initial Model (Thinking-Free) | 82.5 | 1.3 | - | 0.21 |
> | TFPI Stage 1 (Thinking) | 94.4 | 4.8 | 25.0 | 0.21 |
> | TFPI Stage 1 (Thinking-Free) | 88.3 | 1.1 | | 0.91 |
> | TFPI Stage 2 (Thinking) | 94.9 | 4.7 | 26.0 | 0.22 |
> | TFPI Stage 2 (Thinking-Free) | 93.6 | 1.6 | - | 4.0 |
> | TFPI Stage 3 (Thinking) | 94.9 | 5.1 | 30.9 | 0.26 |
> | TFPI Stage 3 (Thinking-Free) | 94.6 | 2.7 | - | 27.0 |
>
> From the table, our previous conclusion also holds for the simpler test set like MATH500.
>
>
> **We also report "reflection word counts" for the table in our response to Question 1**:
>
> | Model | AIME24 Thinking | AIME24 Answer | AIME25 Thinking | AIME25 Answer | Beyond AIME Thinking | Beyond AIME Answer | GPQA Thinking | GPQA Answer |
> | - | - | - | - | - | - | - | - | - |
> | Initial Model (Thinking) | 85.4 | 15.4 | 104.2 | 14.5 | 88.9 | 2.0 | 51.0 | 0.50 |
> | Initial Model (Thinking-Free) | - | 1.29 | - | 1.28 | - | 1.3 | - | 2.6 |
> | TFPI Stage 1 (Thinking) | 61.7 | 7.0 | 74.4 | 7.1 | 62.8 | 0.30 | 34.7 | 0.1 |
> | TFPI Stage 1 (Thinking-Free) | - | 4.8 | - | 5.0 | - | 4.9 | - | 1.6 |
> | TFPI Stage 2 (Thinking) | 65.5 | 8.1 | 76.8 | 4.1 | 70.3 | 0.29 | 40.3 | 0.16 |
> | TFPI Stage 2 (Thinking-Free) | - | 18.3 | - | 20.9 | - |17.0 | - | 6.2 |
> | TFPI Stage 3 (Thinking) | 80.2 | 0.89 | 100.6 | 0.20 | 89.8 | 0.43 | 51.1 | 0.19 |
> | TFPI Stage 3 (Thinking-Free) | - | 37.1 | - | 46.1 | - | 42.7 | - | 14.7 |
>
> From the table, **thinking-free training does not shift part of the reasoning process into the answer segment after training (in thinking-mode inference)**.
>
> We hope these additional results could help resolve your concerns. At this stage, we welcome any follow-up questions or comments and are available to discuss further.
>
> Sincerely,
>
> TFPI Authors

---

### Official Review · Reviewer_vwux · 2025-10-31

**Soundness:** 2
**Presentation:** 3
**Contribution:** 2
**Rating:** 2
**Confidence:** 4

**Summary:**

This paper introduces Thinking-Free Policy Initialization (TFPI), a training method for large reasoning models designed to improve performance and efficiency while reducing computational costs. The core idea here is to use a "Thinking-Free" operation during a multistage Reinforcement Learning process. This is achieved by modifying the input template to explicitly omit the model's chain-of-thought tokens (`</think>` append). The training follows a curriculum (CL) where the context length is progressively increased. The authors conduct experiments on various model sizes and benchmarks, demonstrating that TFPI can achieve strong reasoning performance with significantly less training compute compared to direct long context (includes CoT tokens) RL training.

**Strengths:**

- The paper tackles the significant and practical challenge of the high computational costs associated with training large reasoning models (LRMs) using long-context Reinforcement Learning with Verifiable Reward (RLVR).
- The authors provide an extensive evaluation across multiple model sizes (1.5B, 4B, 7B) and a diverse set of reasoning benchmarks, including mathematics (AIME), multi-task reasoning (GPQA), code generation (LiveCodeBench), and instruction following (IFEval). The inclusion of both in-domain and out-of-domain tasks is noteworthy.
- The results presented are strong and demonstrate high efficiency. For example, the paper reports that a 4B model trained with TFPI achieves 89.0% accuracy on AIME24 using less than 4,000 H20 GPU hours, which is good.
- The paper includes both behavioral (verification analysis) and parameter-level (PCA) analyses to provide insight into why TFPI is effective. This adds value to the empirical results and helps to understand the underlying mechanics of the proposed method.

**Weaknesses:**

The primary weaknesses of this paper relate to the experimental design and the isolation of the core contribution, which makes it difficult to definitively attribute the observed gains to the proposed method.

- The most significant weakness is the absence of a crucial baseline. The paper compares TFPI (multi-stage RL with thinking-free templates) against "Direct RL" (single-stage RL with a long context). However, it fails to compare against a model trained with the **same multi-stage schedule ($2K \rightarrow 4K \rightarrow 8K$) but using the standard "thinking" template**. Without this ablation, it is impossible to disentangle the performance gains and determine whether they stem from the proposed "Thinking-Free" template modification or simply from the known benefits of a multi-stage, curriculum-based training schedule. The improvements could be an artifact of the training curriculum rather than the novelty of TFPI. Clearly, ablations studies are needed.

- The paper acknowledges that multi-stage RLVR (starting with short contexts and gradually increasing them) is a "common mitigation strategy". The core technical contribution thus appears to be the addition of a simple template modification to this existing paradigm. While combining existing ideas can be effective, the lack of the aforementioned ablation makes it difficult to assess the true novelty and impact of the "Thinking-Free" operation itself.

- The paper claims that TFPI exhibits generalizability across domains, even when trained exclusively on mathematics. However, the results show inconsistencies that weaken this claim. For example:

    - In Table 1, the performance of the DS-7B model on GPQA drops from 49.0% after Stage 1 to 46.8% after Stage 3.
    - For the DS-1.5B model, performance on LiveCodeBench fluctuates significantly across stages (18.4% $\rightarrow$ 20.5% $\rightarrow$ 19.9%).
        These fluctuations suggest that the out-of-domain improvements are not consistently monotonic and may be sensitive to the training stage.

- The primary comparison is between TFPI's multi-stage strategy and a single-stage "Direct RL" approach. Even with matched total compute, these are fundamentally different training strategies, making a direct comparison challenging to interpret. Furthermore, evaluation settings are inconsistent, with different max sequence lengths (32K vs 48K) and different decoding parameters for "thinking" vs. "thinking-free" modes, which complicates a fair assessment.

**Questions:**

1. The paper's central claim hinges on the efficiency of the 'Thinking-Free' template. Could the authors conduct the critical ablation experiment: train a model using the exact same multi-stage curriculum ($2K \rightarrow 4K \rightarrow 8K$) but with the standard 'thinking' template enabled throughout? Please provide a direct comparison of both the final performance and the computational resources (e.g., GPU hours and memory usage) consumed against the TFPI-trained models. This is essential to determine if the gains come from the template modification or simply the curriculum.

2. How do the authors explain the performance fluctuations and, in some cases, degradations observed in out-of-domain tasks (e.g., GPQA for DS-7B) in the later stages of TFPI?
3. Could the authors elaborate on the rationale for comparing the multi-stage TFPI against a single-stage "Direct RL" baseline, rather than a multi-stage "thinking" baseline, which would serve as a more direct control for the curriculum aspect of the training?
4. Your evaluation uses inconsistent decoding parameters (such as lower `top-p` for thinking-free) and max sequence lengths (32K vs. 48K) across different models and modes. This introduces confounding variables. How can you demonstrate that the observed efficiency gains are a direct result of the TFPI training, and not an artifact of these different evaluation settings?

---

> ### Author Response · Authors · 2025-11-20
> **Response to Reviewer vwux [1/3]**
>
> Dear Reviewer vwux,
>
> Thank you for the time and effort in reviewing our work. For your concerns, we will address one by one as follows:
>
> ## Weakness 1
> > The most significant weakness is the absence of a crucial baseline. The paper compares TFPI (multi-stage RL with thinking-free templates) against "Direct RL" (single-stage RL with a long context). However, it fails to compare against a model trained with the same multi-stage schedule ($2K \rightarrow 4K \rightarrow 8K$) but using the standard "thinking" template. Without this ablation, it is impossible to disentangle the performance gains and determine whether they stem from the proposed "Thinking-Free" template modification or simply from the known benefits of a multi-stage, curriculum-based training schedule. The improvements could be an artifact of the training curriculum rather than the novelty of TFPI. Clearly, ablations studies are needed.
>
> Thank you for your comments! **As discussed in Section 3.2 and Figure 2 (right), applying Direct RL with a 4K response length causes a substantial performance drop on AIME25 for Qwen3-4B.
> In contrast, TFPI trains stably even at 4K.
> This indicates that, even in a multi-stage setting, the initial stage length should not be too short.**
> We believe that **enabling multi-stage training with short response lengths is a benefit of TFPI**.
> **As suggested, we also added strict ablation with the same stage schedule using Qwen3-4B in Section 5.4 and Appendix F.7. For your convenience, we present the results here**:
>
> |Method|AIME24|AIME25|Beyond AIME|GPQA|LiveCodeBench|IFEval|Overall|
> |-|-|-|-|-|-|-|-|
> |TFPI stage 1 | 75.2 | 67.8 | 42.4 | 57.9 | 55.3 | 66.0 | 60.8 |
> | Multi-Stage Direct RL Stage 1 | 49.2 | 36.0 | 22.6 | 51.4 | 44.0 | 64.5 | 44.6 |
> |TFPI stage 2 | 76.0 | 68.2 | 44.7 | 57.8 | 54.8 | 64.8 | 61.0 |
> | Multi-Stage Direct RL Stage 2 | 58.4 | 42.3 | 28.2 | 52.9 | 48.4 | 63.2 | 48.9 |
> |TFPI stage 3 | 79.9 | 70.6 | 46.7 | 58.5 | 57.0 | 70.2 | 63.8 |
> | Multi-Stage Direct RL Stage 3 | 64.9 | 48.6 | 32.5 | 55.2 | 51.0 | 65.2 | 52.9 |
>
> **We also provide the comparison of training compute in the following table**:
>
> | Training Steps | Training Time (kh) | MFU (\%) |
> |-|-|-|
> | TFPI Stage 1 | 4K, 100 steps | 0.35 | 66.28 |
> | Multi-Stage Direct RL Stage 1 | 4K, 100 steps | 0.45 | 66.82 |
> | TFPI Stage 2 | 8K, 56 steps | 0.52 | 66.67 |
> | Multi-Stage Direct RL Stage 2 | 8K, 56 steps | 0.51 | 66.31 |
> | TFPI Stage 3 | 16K, 64 steps | 0.91 | 66.76 |
> | Multi-Stage Direct RL Stage 3 | 16K, 64 steps | 0.97 | 65.99 |
> | TFPI Total | -  | 1.79 | - |
> | Multi-Stage Direct RL Total | - | 1.93 | - |
>
> Taken together, the tables show that Direct RL with an extremely short context length induces an irreversible performance degradation that later stages fail to recover.
>
> ## Weakness 2
> > The paper acknowledges that multi-stage RLVR (starting with short contexts and gradually increasing them) is a "common mitigation strategy". The core technical contribution thus appears to be the addition of a simple template modification to this existing paradigm. While combining existing ideas can be effective, the lack of the aforementioned ablation makes it difficult to assess the true novelty and impact of the "Thinking-Free" operation itself.
>
> Thank you for your concern. Please refer to our response to Weakness 1.
>
> ## Weakness 3
> > The paper claims that TFPI exhibits generalizability across domains, even when trained exclusively on mathematics. However, the results show inconsistencies that weaken this claim.
>
> Thank you for your careful inspection! Following Reviewer dBLU’s writing suggestion 2 (`Given modest but inconsistent gains (without provided statistical significance) outside math, I would call it some transfer rather than generalizes across domains.`), Our original phrasing could be made more precise: we **have updated the wording from “generalization” to “some transfer beyond mathematics,”** which more accurately reflects our findings.
>
> Interestingly, these fluctuations arise after TFPI surpasses direct RL, indicating that the models quickly reach a performance ceiling on OOD datasets and then oscillate around a convergence point.
>
> As discussed in Section 4.2, these fluctuations suggest that incorporating more diverse training data spanning multiple domains could further stabilize and enhance the performance of TFPI beyond mathematics, which we see as a promising direction for future work.

---

> ### Author Response · Authors · 2025-11-20
> **Response to Reviewer vwux [2/3]**
>
> ## Weakness 4
> > The primary comparison is between TFPI's multi-stage strategy and a single-stage "Direct RL" approach. Even with matched total compute, these are fundamentally different training strategies, making a direct comparison challenging to interpret. Furthermore, evaluation settings are inconsistent, with different max sequence lengths (32K vs 48K) and different decoding parameters for "thinking" vs. "thinking-free" modes, which complicates a fair assessment.
>
> Thank you for your comments! About the multi-stage ablation, please refer to our response to Weakness 1 and Weakness 2.
>
> About the evaluation setting, **we would like to clarify that our comparison is fair for each (initial model, inference mode) combination**.
> Our hyperparameters are consistent within each (initial model, inference mode) combination across all experiments (all tables/figures) throughout the paper. We follow the official recommendations where available, which differ by both base model and mode (thinking vs. thinking‑free). The settings used throughout the paper are:
>
> | Initial Model / Inference Mode | Thinking mode | Thinking-free mode |
> | --- | --- | --- |
> | Qwen3‑4B (and RL‑tuned variants) | T = 0.6, top‑p = 0.95, top‑k = 20, max tokens = 48k | T = 0.7, top‑p = 0.8, top‑k = 20, max tokens = 32k |
> | DeepSeek‑R1‑Distill‑Qwen‑1.5B/7B (and RL‑tuned variants) | T = 0.6, top‑p = 0.95, top‑k = −1 (no top‑k), max tokens = 32k | T = 0.7, top‑p = 0.8, top‑k = 20, max tokens = 32k |
>
> **The reasons are**:
> - Qwen3 series: We adopt the official Hugging Face recommendations, which specify distinct settings for thinking vs. thinking‑free modes: https://huggingface.co/Qwen/Qwen3-4B
> - DeepSeek‑Distill series: The official page (https://huggingface.co/deepseek-ai/DeepSeek-R1-Distill-Qwen-1.5B) specifies thinking‑mode settings (T = 0.6, top‑p = 0.95); we use the default top‑k (−1). As no thinking‑free guidance is provided, we follow the Qwen3 thinking‑free settings.
> - Maximum output length: We use 32k by default. The only exception is Qwen3‑4B in thinking mode. Polaris [1] observes that longer contexts with RoPE scaling help on harder questions and uses 90K for evaluation. We find that 90K is extremely slow for evaluation, and 48K is enough based on the following analysis of Qwen3-4B on AIME25 with varying response lengths:
>
> |Response Length | 16K | 32K | 48K | 64K | 80K | 90K |
> | - | - | - | - | - | - | - |
> |Avg@32 | 42.6 | 64.1 | 68.3 | 68.5 | 68.5 | 68.5 |
> | Clip Ratio | 55.7 | 25.7 | 5.2 | 1.2 | 0.6 | 0.0 |
>
>
> Furthermore, **we also report additional results under a unified configuration (T = 0.6, top‑p = 0.95, top‑k = 20, max tokens = 32k) in Appendix F.4 in our updated PDF**. All main conclusions remain unchanged under this setting.
>
> ## Question 1
> > The paper's central claim hinges on the efficiency of the 'Thinking-Free' template. Could the authors conduct the critical ablation experiment: train a model using the exact same multi-stage curriculum (2K->4K->8K) but with the standard 'thinking' template enabled throughout? Please provide a direct comparison of both the final performance and the computational resources (e.g., GPU hours and memory usage) consumed against the TFPI-trained models. This is essential to determine if the gains come from the template modification or simply the curriculum.
>
> Thanks for the question! Please refer to our response to Weakness 1.
>
> ## Question 2
>
> > How do the authors explain the performance fluctuations and, in some cases, degradations observed in out-of-domain tasks (e.g., GPQA for DS-7B) in the later stages of TFPI?
>
> Thank you for the thoughtful question. We hypothesize that early stages appear to build general capabilities (e.g., avoiding truncation), while subsequent stages with longer contexts bias the model toward longer responses. Many out-of-domain (OOD) tasks, such as GPQA, typically do not require such long responses as AIME, so this specialization can yield some fluctuations.
>
> Another factor might be distribution and format mismatch: OOD benchmarks differ from the training distribution both in domain and in answer format (e.g., boxed numeric answers for math vs. multiple-choice for GPQA or executable code for LiveCodeBench).
> This mismatch makes transfer more sensitive in later stages.
>
> Interestingly, these fluctuations arise after TFPI surpasses direct RL, indicating that the models quickly reach a performance ceiling on OOD datasets and then oscillate around a "convergence point".
>
> As discussed in Section 4.2, these observations suggest that incorporating more diverse, multi-domain, and multi-format training data could help stabilize and further improve TFPI’s performance beyond mathematics.

---

> ### Author Response · Authors · 2025-11-20
> **Response to Reviewer vwux [3/3]**
>
> ## Question 3
>
> > Could the authors elaborate on the rationale for comparing the multi-stage TFPI against a single-stage "Direct RL" baseline, rather than a multi-stage "thinking" baseline, which would serve as a more direct control for the curriculum aspect of the training?
>
> Thank you for the opportunity to clarify. We have taken care to ensure a fair comparison.
>
> As noted in our response to Weakness 1, a multi-stage “thinking” baseline using the identical stage schedule suffers a substantial performance drop. To avoid such a scenario, we evaluate it in a regime that guarantees stable improvement.
>
> We also considered comparing single-stage TFPI (at say 2K/4K) directly against Direct RL. However, TFPI at 2K/4K converges rapidly and uses little training compute; matching compute would afford Direct RL only a small number of steps, yielding an uninformative comparison.
>
> For these reasons, and under matched compute wherever feasible, we believe the present experimental setup provides a fair and informative comparison.
>
> ## Question 4
>
> > Your evaluation uses inconsistent decoding parameters (such as lower top-p for thinking-free) and max sequence lengths (32K vs. 48K) across different models and modes. This introduces confounding variables. How can you demonstrate that the observed efficiency gains are a direct result of the TFPI training, and not an artifact of these different evaluation settings?
>
> Thank you for your question!
> We would like to clarify that our comparison is fair for each (initial model, inference mode) combination.
> In our response to weakness 4, we have explained why we chose different decoding parameters for different modes/initial models.
> Furthermore, **we also report additional results under a unified configuration (T = 0.6, top‑p = 0.95, top‑k = 20, max tokens = 32k) in Appendix F.4 in our revised PDF**.
>
>
> Thank you again for your careful reading and thoughtful comments. We hope our responses have addressed your concerns and questions. At this stage, we remain available for any further discussion. Please feel free to leave any further questions/comments.
>
> Sincerely,
>
> TFPI Authors
>
> [1] POLARIS: A Post-Training Recipe for Scaling Reinforcement Learning on Advanced Reasoning Models

---

### Official Review · Reviewer_gUP8 · 2025-11-02

**Soundness:** 4
**Presentation:** 3
**Contribution:** 3
**Rating:** 8
**Confidence:** 4

**Summary:**

This paper proposes a fast, low-cost initialization phase for long-CoT RL called Thinking-Free Policy Initialization (TFPI). By eliminating the thinking process during generation, it significantly reduces the training cost of Long CoT RL. The implementation is also relatively straightforward: by simply replacing the Rollout in the existing DAPO method with inference that omits the thinking process, it simultaneously lowers RL computational costs and improves performance.

**Strengths:**

1）By replacing the rollout in the existing DAPO method with thinking-free inference, this work achieves simultaneous reduction of RL computational cost and improvement in performance, which is of great significance to RL training.
2）The experimental results are comprehensive, demonstrating that TFPI not only lowers computational cost but also enhances slow-thinking reasoning ability, with validations conducted across domains and settings. In addition, the experimental section discusses, from both behavioral and parameter perspectives, why TFPI is able to improve slow-thinking reasoning performance.

**Weaknesses:**

1）The description of the proposed method in the main text appears rather brief, making it difficult to grasp all the method details clearly.
2）In the experimental section, no details are provided for the choice of hyper-parameters such as temperature, topp, topk, or maximum output length.

**Questions:**

Why are the hyper-parameter settings inconsistent across comparative experiments? For instance, the hyper-parameters of thinking Mode and thinking-free mode in Table 4 are different.

---

> ### Author Response · Authors · 2025-11-20
> **Response to Reviewer gUP8**
>
> Dear Reviewer gUP8,
>
> Thank you for the time and effort you invested in reviewing our work. We are grateful for your constructive feedback and for recognizing our contributions. For your questions, we will answer one by one as follows:
>
> > The description of the proposed method in the main text appears rather brief, making it difficult to grasp all the method details clearly.
>
> We value your helpful comment! As suggested, we **have made the following revisions**:
>
> + Streamlined Section 2 to focus on DAPO.
> + Added a sentence in Section 3.3 explicitly linking x and x' in our methodology to the previously introduced templates, clarifying their correspondence: "Template 1 and Template 2 showcase the correspondence between $x$ and $x'$."
> + Added a concluding sentence in Section 3.3 highlighting the relationship between multi-stage training and our TFPI, to make the subsequent experimental setup easier to follow: "Note that Heuristics for RLVR (e.g., multi-stage RL, data curricula) can be directly applied to TFPI."
>
> We hope these revisions improve clarity and help readers grasp our method and its experimental settings.
>
> > In the experimental section, no details are provided for the choice of hyper-parameters such as temperature, topp, topk, or maximum output length.
>
> Thank you for pointing this out! In our original version, we wrote "with detailed decoding parameters provided in Appendix C.3." in our experimental section. As suggested, we **now briefly cover the choice of hyperparameters for evaluation in Section 4.1 and cross-reference to Appendix C.3 for further details and Appendix F.4 for further experiments**.
>
> **For your convenience, we give our revision in Section 4.1 below:**
>
> "Following the officially recommended decoding parameters, we set the temperature to 0.6 and top-p to 0.95 for the thinking mode, and $0.7, 0.8$ for the thinking-free mode, respectively.
> The maximum output length is 32k for all configurations, except for the Qwen3-4B thinking mode, where 48k is used to reduce the clipping ratio.
> Further details are provided in Appendix C.3, and the effects of changing decoding parameters are presented in Appendix F.4."
>
> > Why are the hyper-parameter settings inconsistent across comparative experiments? For instance, the hyper-parameters of thinking Mode and thinking-free mode in Table 4 are different.
>
> Thank you for your question! Our hyperparameters are consistent within each (initial model, inference mode) combination across all experiments (all tables/figures) throughout the paper. We follow the official recommendations where available, which differ by both base model and mode (thinking vs. thinking‑free). The settings used throughout the paper are:
>
>
> |Initial Model / Inference Mode|Thinking Mode|Thinking-free Mode|
> |-|-|-|
> |Qwen3‑4B (and RL‑tuned variants)|T=0.6, top‑p=0.95, top‑k=20, max tokens=48k|T=0.7, top‑p=0.8, top‑k=20, max tokens=32k|
> |DeepSeek‑R1‑Distill‑Qwen‑1.5B/7B (and RL‑tuned variants)|T=0.6, top‑p=0.95,t op‑k=−1, max tokens=32k|T=0.7, top‑p=0.8, top‑k=20, max tokens=32k|
>
>
> **The reasons are**:
> - Qwen3 series: We adopt the official Hugging Face recommendations, which specify distinct settings for thinking vs. thinking‑free modes: https://huggingface.co/Qwen/Qwen3-4B
> - DeepSeek‑Distill series: The official page (https://huggingface.co/deepseek-ai/DeepSeek-R1-Distill-Qwen-1.5B) specifies thinking‑mode settings (T = 0.6, top‑p = 0.95); we use the default top‑k (−1). As no thinking‑free guidance is provided, we follow the Qwen3 thinking‑free settings.
> - Maximum output length: We use 32k by default. The only exception is Qwen3‑4B in thinking mode. Polaris [1] observes that longer contexts with RoPE scaling help on harder questions and uses 90K for evaluation. We find that 90K is extremely slow for evaluation, and 48K is enough based on the following analysis of Qwen3-4B on AIME25 with varying response lengths:
>
> |Response Length|16K|32K|48K|64K|80K|90K|
> |-|-|-|-|-|-|-|
> |Avg@32|42.6|64.1|68.3|68.5|68.5|68.5|
> |Clip Ratio|55.7|25.7|5.2|1.2|0.6|0.0|
>
> Furthermore, we **also report additional results under a unified configuration (T = 0.6, top‑p = 0.95, top‑k = 20, max tokens = 32k) in Appendix F.4 in our updated PDF**. All main conclusions remain unchanged under this setting.
>
> For your convenience, we also report the results of our meta-experiment in Figure 2 (left) here (under a unified evaluation setting):
>
> |Model | \# Tokens (Thinking) | \# Tokens (Thinking-Free) | $\Delta$ |
> |-|-|-|-|
> |DS-1.5B (original setting)  | 16.5K | 4.4K | -73\% |
> |Unified Setting | 16.4K | 3.6K | -78\% |
> | \texttt{Qwen3-4B} (original setting)  | 17.4K | 5.1K | -71\% |
> | Unified Setting | 17.4K | 4.7K | -73\% |
>
> Thank you for your valuable comments. We hope our responses have addressed your questions. At this stage, we remain available for any further discussion.
>
> Sincerely,
>
> TFPI Authors
>
> [1] POLARIS: A Post-Training Recipe for Scaling Reinforcement Learning on Advanced Reasoning Models

---

### Official Review · Reviewer_dBLU · 2025-11-03

**Soundness:** 3
**Presentation:** 3
**Contribution:** 3
**Rating:** 8
**Confidence:** 4

**Summary:**

The paper proposes Thinking-Free Policy Initialization (TFPI), a short, low-cost RLVR stage that trains an SFT-distilled long-CoT model using `thinking-free' prompts (explicitly closing a <think> section with </think>). The hypothesis is that (i) thinking-free rollouts dramatically cut sequence length and compute; (ii) training the policy on this setting nonetheless improves slow-thinking performance when evaluated with normal CoT; and (iii) TFPI provides a stronger starting point for subsequent long-context RLVR, raising final performance and further improving token efficiency. Experiments across across math datasets, GPQA, code (LiveCodeBench) and instruction following (IFEval) support these claims.

**Strengths:**

1. The paper proposes a simple and practical method that shows promising results both in terms of training efficiency and performance.
2. Under similar compute budgets, TFPI improves slow-thinking performance while yielding shorter outputs in thinking-free inference.
3. The paper shows several useful ablations and analyses, and compares to several baselines.

**Weaknesses:**

**I do not find any major weaknesses in the paper.** Nonetheless some aspects of the paper could be improved:


1. Table 5 in Appendix shows that the full RL method is trained only for 20 steps, in order to match the compute with the baselines. However, it isn't clear whether the same would hold if the compute for all methods increased? Therefore it remains unclear, whether TFPI only works in low train compute settings, or even if the goal is to squeeze last bit of performance from RL training?
2. The paper uses Polaris-53K dataset, however given they had an extensive study on optimal temperature for RL showcasing the optimal temperature to be 1.4 for Qwen3-4B, why did authors choose to use 1.0? This might also mean, that performance of direct RL could be further improved by just changing this hyperparameter (though it might benefit TFPI as well).
3. PCA Analysis: Distances in low-dimensional projections can be misleading in high-dimensional weight spaces. The cosine similarity plot (alignment with the direct RL direction) is suggestive, but adding baselines (e.g., alignment to a separate training run, or to the early direct-RL update) would make the analysis more robust. Because if both TFPI and direct RL share the same reward and data, some alignment is expected.


Some Writing Suggestions:

1. The matched compute is using wall-clock, but that alone doesn't give full picture, since full utilization of gpu may not be happeining. So providing other variables such as total processed tokens, rl steps (already provided in appendix), etc would further add value.
2. `Generalizes across domains even when trained solely on mathematics.` Given modest but inconsistent gains (without provided statistical significance) outside math, I would call it `some transfer` rather than `generalizes across domains.`.
3. Please bold the numbers in all relevant tables (such as Table 1, 2)
4. I think parts of the paper could be simplified, equations for PPO and GRPO might be unecessary, or could be moved to appendix.

**Questions:**

1. Stage schedule sensitivity: Only (2K->4K->8K) and (4K->8K->16K) are tested. Do different schedules (e.g., more stages, skipping a stage, starting at 8K) matter? Is TFPI effective if started at 8K or 16K (eg, for hard datasets even thinkfree could lead to more than 8K tokens with Qwen3 models)?

---

> ### Author Response · Authors · 2025-11-20
> **Response to Reviewer dBLU [1/2]**
>
> Dear Reviewer dBLU,
>
> Thank you for the time and effort in reviewing our work! We are grateful for your constructive feedback and the recognition of our work.  For your suggestions and comments, we will respond one by one as follows:
>
> ## Suggestion 1
> > Table 5 in Appendix shows that the full RL method is trained only for 20 steps, in order to match the compute with the baselines. However, it isn't clear whether the same would hold if the compute for all methods increased? Therefore it remains unclear, whether TFPI only works in low train compute settings, or even if the goal is to squeeze last bit of performance from RL training?
>
> Thank you for this insightful question! TFPI enables training in a small context length, and performance will finally converge within each stage. If we want to further increase the training compute, the response length could be increased as well, e.g., 24K or 32K for Qwen3-4B. In such a case, we can directly switch to "RL with thinking mode".  We have made a discussion in Section 4.2 (Figure 1 left and Table 2) and Section 5.2 (Figure 5 right). In summary, TFPI accelerates early training and raises the eventual performance ceiling achievable by subsequent conventional RL (more training compute).
>
> ## Suggestion 2
> > The paper uses Polaris-53K dataset, however given they had an extensive study on optimal temperature for RL showcasing the optimal temperature to be 1.4 for Qwen3-4B, why did authors choose to use 1.0? This might also mean, that performance of direct RL could be further improved by just changing this hyperparameter (though it might benefit TFPI as well).
>
> We appreciate your professional comments! We were aware of Polaris’s recommendation of a 1.4 training temperature for Qwen3-4B and therefore ran some experiments to assess its impact in our setup before our main experiments. We ultimately used 1.0 because it delivered comparable accuracy with better rollout stability and readability. **We have added the corresponding discussion in  Appendix F.5 of our revised manuscript.** For your convenience, we breif the reasons here as well:
>
> - Accuracy vs. temperature (Qwen3-4B, AVG@32 on AIME24/25). We observe a broad plateau from 0.8 to 1.4 with very similar performance; only at 1.8 does accuracy clearly degrade.
>
> | Temperature | 0.2 | 0.4 | 0.6 | 0.8 | 1.0 | 1.2 | 1.4 | 1.6 | 1.8 |
> | - | - | - | - | - | - | - | - | - | - |
> | AIME24 (%) | 70.5 | 71.8 | 73.4 | 73.5 | 74.6 | 73.1 | 74.5 | 72.1 | 50.4 |
> | AIME25 (%) | 63.5 | 64.6 | 63.5 | 65.8 | 65.6 | 65.8 | 66.1 | 64.3 | 44.0 |
> - Rollout quality: **Training with 1.4 produced noticeably more garbled/off-format text compared to 1.0, indicating higher lexical diversity at the cost of reasoning-trace readability**.
> - Direct comparison at 1.0 vs. 1.4: **For both TFPI Stage 1 and Direct RL, we trained with temperature 1.0 and 1.4 and observed similar AIME25 outcomes (see Fig. 7 in Appendix F.5 of our revised manuscript).**
>
> Therefore, we lean towards a default training temperature (1.0) for all our experiments.
>
> ## Suggestion 3
> > PCA Analysis: Distances in low-dimensional projections can be misleading in high-dimensional weight spaces. The cosine similarity plot (alignment with the direct RL direction) is suggestive, but adding baselines (e.g., alignment to a separate training run, or to the early direct-RL update) would make the analysis more robust. Because if both TFPI and direct RL share the same reward and data, some alignment is expected.
>
> Thank you for your thoughtful comments! As recommended, we **have added a baseline showing alignment to an early Direct-RL update (Appendix F.1) and included a cross-reference in Section 5.1 (see our update PDF)**.

---

> ### Author Response · Authors · 2025-11-20
> **Response to Reviewer dBLU [2/2]**
>
> ## Question 1
> > Stage schedule sensitivity: Only (2K->4K->8K) and (4K->8K->16K) are tested. Do different schedules (e.g., more stages, skipping a stage, starting at 8K) matter? Is TFPI effective if started at 8K or 16K (eg, for hard datasets even thinkfree could lead to more than 8K tokens with Qwen3 models)?
>
> We appreciate this helpful question. We add experiments using 8K->16K; 16K only with Qwen3-4B, the results are shown below:
>
> |Schedule|AIME24|AIME25|Beyond AIME|GPQA|LiveCodeBench|IFEval|Overall|
> |-|-|-|-|-|-|-|-|
> | Initial Model | 73.6 | 68.3 | 43.4 | 56.8 | 54.9 | 64.9 | 60.3 |
> | Direct RL | 75.7 | 67.0 | 43.6 | 56.3 | 52.5 | 66.0 | 60.2 |
> |$4\rm{K}\rightarrow8\rm{K}\rightarrow16\rm{K}$ | **79.9** | 70.6 | **46.7** | 58.5 | **57.0** | **70.2** | **63.8** |
> | $8\rm{K}\rightarrow16\rm{K}$ | 78.0 | 70.2 | **46.7** | **59.5** | 55.8 | 64.5 | 62.4 |
> | $16\rm{K}$ only | 77.1 | **71.5** | 45.8 | 56.4 | 54.4 | 66.0 | 61.9 |
>
> From the table, we have
> + All schedules outperform the Direct RL baseline and achieve comparable performance on math benchmarks, indicating that TFPI is robust to the choice of schedule.
> + 4K->8K->16K yields the best overall results (63.8\% vs. 62.4\% and 61.9\%).
>
> We hypothesize that this stems from an implicit data curriculum induced by dynamic sampling: beginning at 4K improves training efficiency and emphasizes easier problems (1/8-7/8 correctness under 4K context length) early on.
>
> We have **incorporated this table and analysis in Section 5.4 in our revised manuscript**.
>
> ## Writing Suggestions
>
> We really appreciate your constructive writing suggestions to help us improve the quality of our manuscript! We have **made the following revisions accordingly**:
>
> 1. Added Table 6 reporting total generated tokens and average MFU, with accompanying discussion in Appendix C.1.
> 2. Revised our phrasing from “generalizes across domains” to “shows some transfer beyond mathematics” in Section 4.2 and the final paragraph of the introduction.
> 3. Highlighted key numbers in Tables 1–3 in the main text using different colors.
> 4. Made our writing more concise by moving the PPO and GRPO loss to Appendix A.1, while moving the DAPO loss in Section 2.
>
> Thank you for your valuable comments. We have integrated your suggestions throughout the revision to improve the quality and presentation! At this stage, we are happy to engage further and appreciate any additional questions, comments, or suggestions you may have.
>
> Sincerely,
>
> TFPI Authors

---

### Meta-Review · Area_Chair_dzxA · 2026-01-06

**Summary:**

The meta reviewer thinks the main idea of this paper is quite unintuitive and surprising that it would work, the explanation in section 5 is also very high level and didn't providing a convincing study / evidence. However, given that the improvements are real and significant, and the reviewers are more on the positive side, and the major concern from vwux is answered with a strong evidence. I would respect the evaluation of the reviewers and recommend accept for this paper.

**Reviewer Concerns:**

Most of the concerns are resolved by the rebuttal.
- Lack of details (gUP8) resolved.
- Is the improvement coming from multistage instead of thinking free template? (vwux) resolved with a strong evidence is provided that the improvement comes from the template.
- why the specific stage strategy (dBLU, Z381) resolved.

**Reviewer Scores:**

The final score is (8862), there's no further discussion from vwux, given most questions are answered with supporting evidence, I think vwux would raise their score if fully participated.

---

### Decision · Program_Chairs · 2026-01-26

Accept (Poster)